# Improving Generalization of Complex Models under Unbounded Loss Using PAC-Bayes Bounds

**Xitong Zhang**                                                                          *zhangxit@msu.edu*
*Department of Computational Mathematics, Science and Engineering*
*Michigan State University*

**Avrajit Ghosh**                                                                          *ghoshavr@msu.edu*
*Department of Computational Mathematics, Science and Engineering*
*Michigan State University*

**Guangliang Liu**                                                                          *liuguan5@msu.edu*
*Department of Computer Science and Engineering*
*Michigan State University*

**Rongrong Wang**                                                                          *wangron6@msu.edu*
*Department of Computational Mathematics, Science and Engineering*
*Department of Mathematics*
*Michigan State University*

**Reviewed on OpenReview:** *https://openreview.net/forum?id=MP8bmxuWt6*

## Abstract

Previous research on PAC-Bayes learning theory has focused extensively on establishing tight upper bounds for test errors. A recently proposed training procedure called *PAC-Bayes training*, updates the model toward minimizing these bounds. Although this approach is theoretically sound, in practice, it has not achieved a test error as low as those obtained by empirical risk minimization (ERM) with carefully tuned regularization hyperparameters. Additionally, existing PAC-Bayes training algorithms (e.g., Pérez-Ortiz et al. (2021)) often require bounded loss functions and may need a search over priors with additional datasets, which limits their broader applicability. In this paper, we introduce a new PAC-Bayes training algorithm with improved performance and reduced reliance on prior tuning. This is achieved by establishing a new PAC-Bayes bound for unbounded loss and a theoretically grounded approach that involves jointly training the prior and posterior using the same dataset. Our comprehensive evaluations across various classification tasks and neural network architectures demonstrate that the proposed method not only outperforms existing PAC-Bayes training algorithms but also approximately matches the test accuracy of ERM that is optimized by SGD/Adam using various regularization methods with optimal hyperparameters.

## 1 Introduction

The PAC-Bayes bound plays a vital role in assessing generalization by estimating the upper limits of test errors without using validation data. It provides essential insights into the generalization ability of trained models and offers theoretical backing for practical training algorithms (Shawe-Taylor & Williamson, 1997). For example, PAC-Bayes bounds highlight the discrepancy between the training and generalization errors, indicating the need to incorporate regularizers in empirical risk minimization and explaining how larger datasets contribute to improved generalization. Furthermore, the effectiveness of the PAC-Bayes bounds in estimating the generalization capabilities of machine learning models has been supported by extensive experiments across different generalization metrics (Jiang et al., 2019).

Traditionally, PAC-Bayes bounds have been primarily used for quality assurance or model selection (McAllester, 1998; 1999; Herbrich & Graepel, 2000), particularly with smaller machine learning models. Recent work has introduced a framework that minimizes a PAC-Bayes bound during training large neural networks (Dziugaite & Roy, 2017). Ideally, the generalization performance of deep neural networks could be enhanced by directly minimizing its quantitative upper bounds, specifically the PAC-Bayes bounds, without incorporating any other regularization tricks. However, the effectiveness of applying PAC-Bayes training to deep neural networks is challenged by the well-known issue that PAC-Bayes bounds can become vacuous in highly over-parameterized settings (Livni & Moran, 2020). Additionally, selecting a suitable prior, which should be independent of training samples, is critical yet challenging. This often leads to conducting a parameter search for the prior using separate datasets (Pérez-Ortiz et al., 2021). Furthermore, existing PAC-Bayes training methods are typically tailored for bounded loss (Dziugaite & Roy, 2017; 2018; Pérez-Ortiz et al., 2021), limiting their straightforward application to popular losses like Cross-Entropy.

On the other hand, the prevalent training methods for neural networks, which involve minimizing empirical risk with SGD/Adam, achieve satisfactory test performance. However, they often require integration with various regularization techniques to optimize generalization performance. For instance, research has shown that factors such as larger learning rates (Cohen et al., 2021; Barrett & Dherin, 2020), momentum (Ghosh et al., 2022; Cattaneo et al., 2023), smaller batch sizes (Lee & Jang, 2022), parameter noise injection (Neelakantan et al., 2015; Orvieto et al., 2022), and batch normalization (Luo et al., 2018) all induce higher degrees of *implicit regularization*, yielding better generalization. Besides, various *explicit regularization* techniques, such as weight decay (Loshchilov & Hutter, 2017), dropout (Wei et al., 2020), label noise (Damian et al., 2021) can also significantly affect generalization. While many studies have explored individual regularization techniques to identify their unique benefits, the interaction among these regularizations remains less understood. As a result, in practical scenarios, one has to extensively tune the hyperparameters corresponding to each regularization technique to obtain the optimal test performance.

Although further investigation is needed to fully understand the underlying mechanisms, training models using ERM with various regularization methods remain the prevalent choice and typically deliver state-of-the-art test performance. While PAC-Bayes training is built upon a solid theoretical basis for analyzing generalization, its wider adoption is limited by existing assumptions about loss and challenges in prior selection. Moreover, it is still an open question regarding how to enhance PAC-Bayes training to match the performance of ERM methods with well-tuned regularizations.

In this paper, we propose a practical PAC-Bayes-bound-based training algorithm that nearly matches the performance of the optimally tuned ERM while being more robust to the choice of hyperparameters. The key differences between our algorithm and the previous ones are:

1. A new PAC-Bayes bound for unbounded loss is adopted for training, providing tighter numerical values for highly over-parametrized models.

2. The PAC-Bayes training is enhanced by an optional second stage of Bayesian training, which uses key parameters estimated from the PAC-Bayes training stage.

We provide mathematical analysis to support the proposed algorithm and conduct extensive numerical experiments to demonstrate its effectiveness.

## 2 Preliminaries

This section outlines the PAC-Bayes framework. For any supervised learning problem, the goal is to find a proper model $\mathbf{h}$ from some hypothesis space $\mathcal{H}$, with the help of the training data $\mathcal{S} \equiv \{z_i\}_{i=1}^m$, where $z_i$ is the training pair with sample $\mathbf{x}_i$ and its label $y_i$. Given the loss function $\ell(\mathbf{h}; z_i) : \mathbf{h} \mapsto \mathbb{R}$, which measures the misfit between the true label $y_i$ and the predicted label by $\mathbf{h}$, the empirical and population/generalization errors are defined as:

$$\ell(\mathbf{h}; \mathcal{S}) = \frac{1}{m} \sum_{i=1}^m \ell(\mathbf{h}; z_i), \quad \ell(\mathbf{h}; \mathcal{D}) = \mathbb{E}_{\mathcal{S} \sim \mathcal{D}}(\ell(\mathbf{h}; \mathcal{S})),$$

by assuming that the training and testing data are both i.i.d. sampled from the same unknown distribution $\mathcal{D}$. PAC-Bayes bounds include a family of upper bounds on the generalization error of the following type.

**Theorem 2.1.** *(Maurer, 2004) Assume the loss function $\ell$ is* **bounded** *within the interval* $[0,1]$. *Given a* **preset** *prior distribution $\mathcal{P}$ over the model space $\mathcal{H}$, and given a scalar $\delta \in (0,1)$, for any choice of i.i.d $m$-sized training dataset $\mathcal{S}$ according to $\mathcal{D}$, and all posterior distributions $\mathcal{Q}$ over $\mathcal{H}$,*

$$\mathbb{E}_{\mathbf{h}\sim\mathcal{Q}}\ell(\mathbf{h};\mathcal{D}) \leq \mathbb{E}_{\mathbf{h}\sim\mathcal{Q}}\ell(\mathbf{h};\mathcal{S}) + \sqrt{\frac{\log(\frac{2\sqrt{m}}{\delta}) + \mathrm{KL}(\mathcal{Q}||\mathcal{P})}{2m}},$$

*holds with probability at least $1 - \delta$. Here, KL stands for the Kullback-Leibler divergence.*

A PAC-Bayes bound measures the gap between the expected empirical and generalization errors. It's worth noting that this bound holds for all posterior $\mathcal{Q}$ for any given data-independent prior $\mathcal{P}$ and, which enables optimization of the bound by searching for the best posterior. In practice, the posterior mean corresponds to the trained model, and the prior mean can be set to the initial model. In this paper, we will use $||\cdot||$ to denote a generic norm, and $||\cdot||_2$ to denote the $L_2$ norm.

**The focus on Neural Networks**: PAC-Bayes training algorithms, including the one proposed here, can be applied to a wide range of supervised learning problems. However, in this paper, we will focus on the training of deep neural networks for the following reasons:

Over-parameterized deep neural networks present a significant challenge for PAC-Bayes training, as the Kullback-Leibler (KL) term in the PAC-Bayes bounds is believed to grow rapidly with the number of parameters, quickly leading to vacuous bounds. According to the current literature, deep networks are indeed one of the most critical models where existing PAC-Bayes training algorithms encounter difficulties.

Intuitively, minimizing the generalization bound does not require additional implicit or explicit regularization, thus reducing the need for tuning. This advantage of reduced tuning is particularly significant in the training of neural networks, where extensive tuning is typically required for ERM. However, for classical problems such as Lasso, which involve only one or two hyperparameters, this advantage of PAC-Bayes training may be less pronounced.

## 3    Related Work

PAC-Bayes bounds were first used to train neural networks in Dziugaite & Roy (2017). Specifically, the bound McAllester (1999) has been employed for training shallow stochastic neural networks on binary MNIST classification with bounded 0-1 loss and has proven to be non-vacuous. Following this work, many recent studies (Zhou et al., 2018; Letarte et al., 2019; Rivasplata et al., 2019; Pérez-Ortiz et al., 2021; Biggs & Guedj, 2021; Perez-Ortiz et al., 2021; Viallard et al., 2023) expanded the applicability of PAC-Bayes bounds to a wider range of neural network architectures and datasets. However, most studies are limited to training shallow networks with binary labels using bounded loss, which restricts their broader application to deep network training. Although PAC-Bayes bounds for unbounded loss have been established (Audibert & Catoni, 2011; Alquier & Guedj, 2018; Holland, 2019; Kuzborskij & Szepesvári, 2019; Haddouche et al., 2021; Rivasplata et al., 2020; Rodríguez-Gálvez et al., 2023; Casado et al., 2024), it remains unclear whether these bounds can lead to enhanced test performance in training neural networks. This uncertainty arises partly because they usually include assumptions that are difficult to validate or terms that are hard to compute in real applications. For example, Kuzborskij & Szepesvári (2019) derived a PAC-Bayes bound under the second-order moment condition of the unbounded loss. However, as mentioned in the paper, that bound is semi-empirical, in the sense that it contains the *population second order moment of the loss with respect to both the posterior and the data distributions.* Since conditional on the posterior, the samples are no longer i.i.d., this type of bound is difficult to estimate. To the best of our knowledge, existing PAC-Bayes bounds built under the second-order moment condition all suffer from this issue.

Recently, Dziugaite et al. (2021) suggested that a tighter PAC-Bayes bound could be achieved with a data-dependent prior. They divide the data into two sets, using one to train the prior and the other to train the posterior with the optimized prior, thus making the prior independent from the training dataset

for the posterior. This, however, reduces the training data available for the posterior. Dziugaite & Roy (2018) and Rivasplata et al. (2020) justified the approach of learning the prior and posterior with the same set of data by utilizing differential privacy. However, the argument only holds for priors provably satisfying the so-called $DP(\epsilon)$-condition in differential privacy, which limits their practical application. Pérez-Ortiz et al. (2021) also empirically shows training with Dziugaite & Roy (2018) could sacrifice test accuracy if the bound is not tight enough. In this work, we advance the PAC-Bayes training approach, enhancing its practicality and showcasing its potential in challenging settings.

# 4 Proposed method

## 4.1 Motivation

To help readers understand the challenges involved in designing practical PAC-Bayes training algorithms for deep neural networks, we begin by offering a detailed examination of the limitations inherent in current PAC-training algorithms and PAC-Bayes bounds. Most popular PAC-Bayes training algorithms (Dziugaite & Roy, 2017; 2018; Pérez-Ortiz et al., 2021) are designed for learning problems with bounded loss. When dealing with unbounded loss, it is necessary to clip the loss to a finite range before training, which can result in suboptimal performance, as demonstrated in Table 1 in the numerical section.

Some PAC-Bayes bounds for unbounded loss have been established in the literature, extending the requirement from bounded loss to sub-Gaussian loss (Theorem 5.1 of Alquier (2021)), loss that satisfies the so-called hypothesis-dependent range (HYPE) condition (Haddouche et al., 2021), and loss controlled by some finite cumulant generating function (CGF) (Rodríguez-Gálvez et al., 2023). More specifically, in (Haddouche et al., 2021), the loss requirement is relaxed to the existence of $K(h)$ such that $sup_{z\sim\mathcal{D}}\ell(h,z) \leq K(h), \forall h \in \mathcal{H}$. This is known as the HYPE condition and is much weaker than the requirement for a bounded loss. However, commonly used cross-entropy loss still does not satisfy this condition without additional assumptions on the boundedness of both the input and the model. Furthermore, our numerical experiments indicate that minimizing bounds tailored for sub-Gaussian and CGF losses during training is also largely ineffective. Specifically, on CIFAR10 with CNN9, the bound values for sub-Gaussian and CGF losses are 5.17 and 5.08, respectively. These values almost do not decrease during training and are even higher than the initial test loss (with a value of 2.30) under random weight initialization, indicating that these bounds can not contribute to improving the training performance.

Lastly, the PAC-Bayes bound established under the (theoretically) weakest assumption is in Kuzborskij & Szepesvári (2019), where the loss is only required to have a finite second-order moment. The associated bound holds with probability $1 - e^x, x \geq 2$:

$$\mathbb{E}_{\mathbf{h}\sim\mathcal{Q}}\ell(\mathbf{h};\mathcal{D}) \leq \mathbb{E}_{\mathbf{h}\sim\mathcal{Q}}\ell(\mathbf{h};\mathcal{S}) + \sqrt{2\left(\frac{1}{n^2} + \beta\right)\left(\text{KL}\left(\mathcal{Q}||\mathcal{P}\right) + x + \frac{x}{2}\ln\left(1 + m^2\beta\right)\right)}, \quad (1)$$

where $\beta = \mathbb{E}_{\mathbf{h}\sim\mathcal{Q}}\left[\ell^2(\mathbf{h};\mathcal{S}) + \mathbb{E}_{z'\sim\mathcal{D}}\ell^2(\mathbf{h};z')\right]$. However, this bound is not easily amendable for training as the term $\mathbb{E}_{\mathbf{h}\sim\mathcal{Q}}\mathbb{E}_{z'\sim\mathcal{D}}\ell^2(\mathbf{h};z')$ in $\beta$ is a *population* second-order moment of the loss with respect to the data and the posterior distribution. Since the data is no longer i.i.d. when conditioned on the posterior, estimating this population's second-order moment becomes challenging.

The above-mentioned limitations in related literature motivate us to design a new PAC-Bayes bound for unbounded loss that is easy to apply in training.

## 4.2 A new PAC-Bayes Bound for Unbounded loss

We propose a training-friendly PAC-Bayes bound that holds under mild conditions.

### 4.2.1   Condition under which the new bound will hold

The new bound we shall propose is based on the following assumption of the loss function in Definition 4.1. Please note that the $X$ in Definition 4.1 will later represent the training loss in the PAC-Bayes analysis, and $\mathbb{E}[X]$ will represent the population loss.

**Definition 4.1** (Exponential moment on finite intervals)**.** Let $X$ be a random variable defined on the probability space $(\Omega, \mathcal{F}, \mathcal{P})$ and $0 \leq \gamma_1 \leq \gamma_2 \leq \infty$ be two numbers. We call any $K > 0$ an exponential moment bound of $X$ over the interval $[\gamma_1, \gamma_2]$, when

$$\mathbb{E}[\exp(\gamma(\mathbb{E}[X] - X))] \leq \exp(\gamma^2 K) \tag{2}$$

holds for all $\gamma \in [\gamma_1, \gamma_2]$.

We provide a few remarks to clarify the position of this definition in the literature.

*Remark* 4.2 (The left-side moment)*.* Similar to certain other PAC-Bayes conditions in the literature (Alquier, 2021), this definition only requires the left-side moment to be bounded. Specifically, it requires $\mathbb{E}[\exp(\gamma(\mathbb{E}[X] - X))]$ to be bounded for $\gamma > 0$, rather than for all $\gamma \in \mathbb{R}$. The need for this left-side condition arises from the goal of establishing an upper bound on the population loss. An upper bound on the population loss $\mathbb{E}[X]$ in terms of the empirical loss $X$ translates to upper bounding $\mathbb{E}[X] - X$. For positive $\gamma$, this leads to the left-side condition.

*Remark* 4.3 ($\gamma$ within a finite interval bounded away from 0)*.* In practice, we set both $\gamma_1$ and $\gamma_2$ to positive values. We observe that restricting $\gamma$ to a finite interval $[\gamma_1, \gamma_2]$ with positive $\gamma_1, \gamma_2$ often reduces the bound $K$. In contrast, all previous bounds use $\gamma_1 = 0$.

*Remark* 4.4 (Non-negative loss)*.* Since most loss functions in machine learning (e.g., Cross-Entropy, $L_1$, MSE, Huber loss, hinge loss, Log-cosh loss, quantile loss) are non-negative, it is of great interest to analyze the strength of Definition 4.1 under $X \geq 0$. In this case, we can show that Definition 4.1 is weaker than the second-order moment condition.

**Lemma 4.5** (Comparison with the second-order-moment condition)**.** *For non-negative random variable $X \geq 0$, the existence of $K$ on the interval $\gamma \in [0, \infty)$ in Definition 4.1 can be implied by the existence of the second-order moment $\mathbb{E}X^2 < \infty$.*

The assumption $X \geq 0$ in Lemma 4.5 can be further relaxed to $X \geq -M$ with $M > 0$, as in this case the random variable $X + M$ is non-negative to which Lemma 4.5 can be applied.

*Remark* 4.6 (Comparison with the first-order-moment condition)*.* Still under the assumption $X \geq 0$, when the $\gamma_1$ in Definition 4.1 is finite (bounded away from 0), the existence of $K$ can be implied by the existence of first-order moment. Indeed, by taking $K = \frac{\mathbb{E}[X]}{\gamma_1}$, the inequality $\mathbb{E}[X] - X \leq \mathbb{E}[X]$ (assumed $X \geq 0$) immediately implies Equation (2). However, this argument does not hold when $\gamma_1 \to 0$. Hence we cannot say Definition 4.1 is as weak as the first-order moment condition.

*Proof of Lemma 4.5.* We show that $\mathbb{E}X^2 < \infty$ implies Definition 4.1 holding for any $\gamma \in [0, \infty)$ with some finite $K$. Since $\mathbb{E}X^2 < \infty$, we have $(\mathbb{E}X)^2 \leq \mathbb{E}X^2 < \infty$. If $\gamma \geq \frac{1}{\mathbb{E}X}$, then it suffices to take the $K$ in

$$\mathbb{E}e^{\gamma(\mathbb{E}X - X)} \leq e^{\gamma^2 K}$$

to be $K = \frac{\mathbb{E}X}{\gamma} \leq (\mathbb{E}X)^2 \equiv K_1$. If $\gamma < \frac{1}{\mathbb{E}X}$, then using the inequality

$$e^x \leq 1 + x + x^2, \quad \forall x < 1$$

with $x := \gamma(\mathbb{E}X - X) \leq \gamma\mathbb{E}X < 1$, we have

$$\mathbb{E}e^{\gamma(\mathbb{E}X - X)} \leq \mathbb{E}(1 + \gamma(\mathbb{E}X - X) + \gamma^2(\mathbb{E}X - X)^2) = 1 + \gamma^2 \text{Var}(X) \leq e^{\gamma^2 \text{Var}(X)}$$

Therefore, it suffices to take $K = \text{Var}(X) \equiv K_2$. Collecting the two cases, we see taking $K = \max\{K_1, K_2\}$ would be enough for Definition 4.1 to hold with $\gamma_1 = 0, \gamma_2 = \infty$. $\qquad\square$

Now, we generalize Definition 4.1 to the PAC-Bayes setting where the random variable is parameterized by models in a hypothesis space.

Let us first explain what we mean by random variables parameterized by models in a hypothesis space. In supervised learning, we define $X(\mathbf{h})$ as $X(\mathbf{h}) \equiv \ell(\mathbf{h}(x), y)$, where $\mathbf{h}$ is the model and $\ell$ is the misfit between the model output $\mathbf{h}(x)$ and the data $y$. For a fixed model $\mathbf{h}$, $X(\mathbf{h})$ is a random variable whose randomness comes from the input pairs $(x, y) \sim \mathcal{D}$ ($\mathcal{D}$ is the data distribution). Since $X(\mathbf{h})$ varies with $\mathbf{h}$, we call it a random variable parameterized by the model $\mathbf{h}$.

**Definition 4.7** (Exponential moment over hypotheses). Let $X(\mathbf{h})$ be a random variable parameterized by the hypothesis $\mathbf{h}$ in some hypothesis space $\mathcal{H}$ (i.e., $\mathbf{h} \in \mathcal{H}$), and fix an interval $[\gamma_1, \gamma_2]$ with $0 \leq \gamma_1 < \gamma_2 \leq \infty$. Let $\{\mathcal{P}_{\boldsymbol{\lambda}}, \boldsymbol{\lambda} \in \Lambda\}$ be a family of distribution over $\mathcal{H}$ parameterized by $\boldsymbol{\lambda} \in \Lambda \subseteq \mathbb{R}^k$. Then, we call any non-negative function $K(\boldsymbol{\lambda})$ an exponential moment bound for $X(\mathbf{h})$ over the priors $\{\mathcal{P}_{\boldsymbol{\lambda}}, \boldsymbol{\lambda} \in \Lambda\}$ and the interval $[\gamma_1, \gamma_2]$, if the following holds

$$\mathbb{E}_{\mathbf{h} \sim \mathcal{P}_{\boldsymbol{\lambda}}} \mathbb{E}[\exp\left(\gamma(\mathbb{E}[X(\mathbf{h})] - X(\mathbf{h}))\right)] \leq \exp\left(\gamma^2 K(\boldsymbol{\lambda})\right),$$

for all $\gamma \in [\gamma_1, \gamma_2]$, and any $\boldsymbol{\lambda} \in \Lambda \subseteq \mathbb{R}^k$. The minimal such $K(\boldsymbol{\lambda})$ is

$$K_{\min}(\boldsymbol{\lambda}) = \sup_{\gamma \in [\gamma_1, \gamma_2]} \frac{1}{\gamma^2} \log(\mathbb{E}_{\mathbf{h} \sim \mathcal{P}_{\boldsymbol{\lambda}}} \mathbb{E}[\exp\left(\gamma(\mathbb{E}[X(\mathbf{h})] - X(\mathbf{h}))\right)]). \tag{3}$$

Definition 4.7 is a direct extension of Definition 4.1 to random variables parametrized by a hypothesis space. Similar to Definition 4.1, when dealing with non-negative loss, the existence of the exponential moment bound $K_{\min}$ is guaranteed, provided that the second-order moment of the loss is bounded, or provided that the first-order moment of the loss is bounded and $\gamma_1$ is bounded away from 0.

*Remark* 4.8 (Dependency of $K$ on prior parameters). The key difference in Definition 4.7 compared to prior work is that we allow the exponential moment bound $K$ to depend on the prior parameter $\boldsymbol{\lambda}$, rather than requiring a single $K$ to hold for all possible $\boldsymbol{\lambda} \in \Lambda$. This advantage carries over into the PAC-Bayes bound established in the next section.

In practice, there are two options to compute $K(\boldsymbol{\lambda})$: one is to use the upper bound derived in the proof of Lemma 4.5, and the other is to estimate $K_{\min}$ directly from the data, via Equation (3). For both options, we need to estimate the expectation of some random variable over the prior distribution $\mathcal{P}_{\boldsymbol{\lambda}}$ for a given $\boldsymbol{\lambda}$ from the training data. Fortunately, this is much easier than estimating the expectation with respect to the posterior distribution $\mathcal{Q}$, as is required by previous work Kuzborskij & Szepesvári (2019), since in our case, after conditioning on the prior, the training data remains i.i.d., allowing a reliable approximation of the population mean by the empirical mean when the data is abundant.

### 4.2.2 A new PAC-Bayes bound

We are ready to present the PAC-Bayes bound for losses that satisfy Definition 4.7.

**Theorem 4.9** (PAC-Bayes bound for unbounded loss with a **preset** prior distribution). *Given a prior distribution $\mathcal{P}_{\boldsymbol{\lambda}}$ over the hypothesis space $\mathcal{H}$, parametrized by $\boldsymbol{\lambda} \in \Lambda$. Assume the loss $\ell(\mathbf{h}, z_i)$ as a random variable parametrized by $\mathbf{h}$ satisfies Definition 4.7. For any $0 < \delta < 1$ and $\gamma \in [\gamma_1, \gamma_2]$, we have*

$$\Pr_{\mathcal{S}}\left(\forall \mathcal{Q} \in \mathbf{Q}, \mathbb{E}_{\boldsymbol{h} \sim \mathcal{Q}} \ell(\mathbf{h}; \mathcal{D}) \leq \mathbb{E}_{\boldsymbol{h} \sim \mathcal{Q}} \ell(\mathbf{h}; \mathcal{S}) + \frac{1}{\gamma m}(\log \frac{1}{\delta} + \mathrm{KL}(\mathcal{Q}||\mathcal{P}_{\boldsymbol{\lambda}})) + \gamma K(\boldsymbol{\lambda})\right) \geq 1 - \delta$$

*where $\mathbf{Q}$ is the set of all probability distributions.*

The proof of this theorem is available in Appendix A.1.

*Remark* 4.10 (Asymptotic convergence rate). By setting $\gamma, \gamma_1 = O(m^{-1/2})$, we observe that the asymptotic behavior of this bound aligns with the $O(m^{-1/2})$ convergence rate of popular PAC-Bayes bounds in the literature.

*Remark* 4.11 (Applicability on CE loss). This theorem, when combined with Lemma 4.5, guarantees that the $O(m^{-1/2})$ convergence rate is achieved for CE loss under a bounded second-order moment condition.

*Remark* 4.12 (The role of $K(\boldsymbol{\lambda})$). The improvement of the new bound primarily stems from our definition of $K(\boldsymbol{\lambda})$. Specifically, Def 4.7 allows $K$ to be dependent on both the range of $\gamma$ and the prior parameter $\lambda$. This flexibility reduces the value of $K$. The trade-off is that $\gamma_1$ and $\gamma_2$ need to be selected in advance as hyperparameters, and $K(\lambda)$ must be estimated from data and optimized during training, which slightly increases the difficulty of the optimization.

To support the claim of Remark 4.12, we provide numerical evidence of the advantage of our $K$ in Figure 1. We use (Rodríguez-Gálvez et al., 2023) and Alquier (2021) as the baseline PAC-Bayes bounds for unbounded loss for comparison. First, observe that both baseline bounds can be written in the form of

$$\mathbb{E}_{\mathbf{h}\sim\mathcal{Q}}\ell(\mathbf{h};\mathcal{D}) \lesssim \mathbb{E}_{\mathbf{h}\sim\mathcal{Q}}\ell(\mathbf{h};\mathcal{S}) + \frac{\mathrm{KL}(\mathcal{Q}||\mathcal{P})}{m\gamma} + \gamma K, \tag{4}$$

where $\lesssim$ hides some absolute constant. Namely, both bounds have a $K$ term, although they follow different definitions. When the loss is assumed to be sub-Gassuain, $K$ is defined as the sub-Gaussian norm of the loss (Theorem 5.1 of Alquier (2021)). Under the CGF condition, $K$ is calculated based on the specific choice of the CGF (Rodríguez-Gálvez et al., 2023) (see Appendix C.2 for our choice of CGF in this experiment). For our bound, the $K$ is chosen as our $K_{\min}(\boldsymbol{\lambda})$ for various choices of $\gamma_1$ and $\gamma_2$. This experiment is conducted on CIFAR10 using CNN9. We follow standard practice in PAC-Bayes training literature by using a Gaussian prior, fixing its mean to the random Kaiming initialization, and picking a universal variance for all the weights. According to Definition 4.7, our $K$ depends on $\gamma_1$ and $\gamma_2$, and the prior parameter $\boldsymbol{\lambda}$, which we set to the universal standard deviation (std) of the prior weight distribution. Since we observed that the variation of $K$ with respect to $\gamma_2$ is very minimal, Figure 1 only plots $K$ as a function of the prior std and $\gamma_1$. The $K$ values in the previous two bounds (i.e., sub-Gaussian and CGF) do not depend on these parameters. Hence, they are represented by two horizontal planes. We observe the following:

- our $K$ values approach those in the CGF bound as $\gamma_1$ approaches 0 and the prior std becomes large;

- our $K$ is significantly smaller than those in the other two bounds near the region of the optimal choice of prior std and $\gamma_1$, which we determined by testing various pairs of $\gamma_1$ and prior standard deviation for the combination that yields the tightest bound after optimized over the posterior. This explains the advantage of making $K$ a function of the prior std;

- since all other terms are the same, the smaller the $K$ is, the tighter the bound is after optimizing over $\gamma$.

- Figure 1 also indicates that the value of $K_{\min}$ is more stable with respect to $\gamma_1$ than with respect to the prior standard deviation, especially near the optimal parameter region, suggesting that choosing $\gamma_1$ is easier. Therefore, in our algorithm, we fix $\gamma_1$ in advance while optimizing over the prior variance $\lambda$ during training. It turns out that the same choice of $\gamma_1$ works for a wide variety of architectures, as shown in the numerical section;

### 4.3   PAC-Bayes training based on the new bound

With the relaxed requirements on the loss function, the proposed bound offers a basis for establishing effective optimization over both the posterior and the prior. We will first outline the training process, which focuses on jointly optimizing the prior and posterior to avoid the complex hyper-parameter search over the prior as Pérez-Ortiz et al. (2021), followed by a discussion of its theoretical guarantees. The procedure is similar to the one in Dziugaite & Roy (2017), but has been adapted to align with our newly proposed bound.

We begin by parameterizing the posterior distribution as $\mathcal{Q}_{\mathbf{w}}$, where $\mathbf{w} \in \mathbb{R}^d$ represents the parameters of the posterior. Next, we parameterize the prior as $\mathcal{P}_{\boldsymbol{\lambda}}$, where $\boldsymbol{\lambda} \in \mathbb{R}^k$. We operate under the assumption that the prior has significantly fewer parameters than the posterior, that is, $k \ll d$; the relevance of this assumption will become apparent upon examining Theorem 4.15. For our PAC-Bayes training, we propose to optimize over all three variables: $\mathbf{w}$, $\gamma$, and $\boldsymbol{\lambda}$:

$$(\hat{\mathbf{w}}, \hat{\gamma}, \hat{\boldsymbol{\lambda}}) = \arg\min_{\mathbf{w},\boldsymbol{\lambda},\gamma\in[\gamma_1,\gamma_2]} L_{PAC}(\mathbf{w}, \gamma, \boldsymbol{\lambda}, \delta), \tag{P}$$

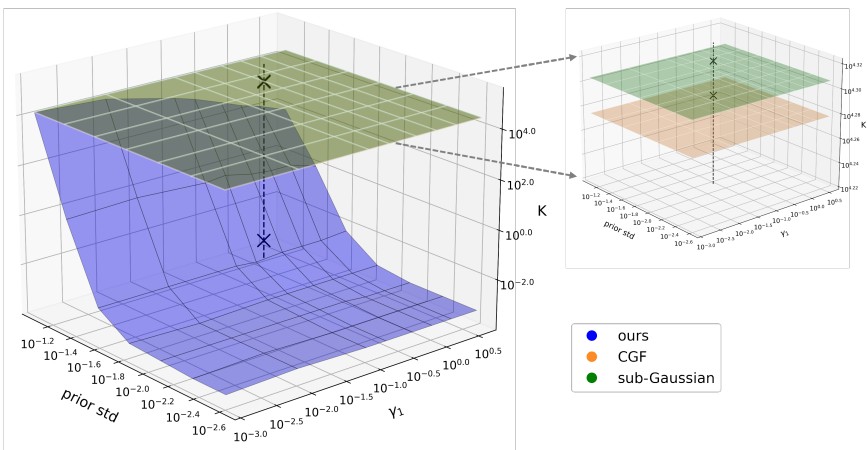

Figure 1: Our definition of $K$ demonstrates a notable advantage in terms of the achieved numerical value compared to the $K$ in the *sub-Gaussian* and *CGF* bounds. Our $K$ varies with $\gamma_1$ and the prior standard deviation, while the previous bounds remain constant as they don't depend on these parameters. The vertical dotted line marks the optimal choice of prior std and $\gamma_1$ during training, which we determined by testing various pairs of $\gamma_1$ and prior standard deviation for the combination that yields the smallest value of the bound after optimized over the posterior. The figure shows that near this optimal combination, our $K$ is much smaller than those in the other two bounds. This experiment is conducted on CIFAR10 using CNN9; see details in Sec. 5.

where

$$L_{PAC}(\mathbf{w}, \gamma, \boldsymbol{\lambda}, \delta) = \mathbb{E}_{\mathbf{h} \sim \mathcal{Q}_{\mathbf{w}}} \ell(\mathbf{h}; \mathcal{S}) + \frac{1}{\gamma m}(\log \frac{1}{\delta} + \mathrm{KL}(\mathcal{Q}_{\mathbf{w}} || \mathcal{P}_{\boldsymbol{\lambda}})) + \gamma K(\boldsymbol{\lambda}). \tag{5}$$

Please note that here $K$ depends on the prior parameter $\boldsymbol{\lambda}$, and we need to optimize it along with other terms. The probability of failure $\delta$ should be fixed in advance.

## 4.4 Theoretical Analysis

Two important questions arise from the training framework based on Equation (5).

1. Is there any theoretical guarantee for the performance of training over the prior?

2. Does the exponential moment bound function $K(\lambda)$ exhibit enough regularity as a function of $\lambda$ to be accurately estimated from the data?

This section addresses these two questions. First, we provide a PAC-Bayes bound that allows the prior to be data-dependent. Then, we validate the Lipschitz continuity of $K(\boldsymbol{\lambda})$ when the prior and posterior are both set to Gaussian distributions.

### 4.4.1 PAC-Bayes bound for arbitrary distribution families

We first make the following assumptions, which will be shown to automatically satisfy by the special Gaussian prior and posteriors.

**Assumption 4.13** (Continuity of the KL divergence)**.** Let $\mathfrak{Q}$ be a family of posterior distributions, let $\mathfrak{P} = \{P_{\boldsymbol{\lambda}}, \boldsymbol{\lambda} \in \Lambda \subseteq \mathbb{R}^k\}$ be a family of prior distributions parameterized by $\boldsymbol{\lambda}$. We say the KL divergence $\mathrm{KL}(\mathcal{Q} || \mathcal{P}_{\boldsymbol{\lambda}})$ is continuous with respect to $\boldsymbol{\lambda}$ over the posterior family, if there exists some non-decreasing function $\eta_1(x) : \mathbb{R}_+ \mapsto \mathbb{R}_+$ with $\eta_1(0) = 0$, such that $|\mathrm{KL}(\mathcal{Q} || \mathcal{P}_{\boldsymbol{\lambda}}) - \mathrm{KL}(\mathcal{Q} || \mathcal{P}_{\tilde{\boldsymbol{\lambda}}})| \leq \eta_1(\|\boldsymbol{\lambda} - \tilde{\boldsymbol{\lambda}}\|)$, for all pairs $\boldsymbol{\lambda}, \tilde{\boldsymbol{\lambda}} \in \Lambda$ and for all $\mathcal{Q} \in \mathfrak{Q}$.

**Assumption 4.14** (Continuity of the $K(\boldsymbol{\lambda})$). Let $K_{\min}(\boldsymbol{\lambda})$ be as defined in Definition 4.7. Assume it is Lipschitz continuous with respect to the parameter $\boldsymbol{\lambda}$ of the prior in the sense that there exists a non-decreasing function $\eta_2(x) : \mathbb{R}_+ \mapsto \mathbb{R}_+$ with $\eta_2(0) = 0$ such that $|K_{\min}(\boldsymbol{\lambda}) - K_{\min}(\tilde{\boldsymbol{\lambda}})| \leq \eta_2(\|\boldsymbol{\lambda} - \tilde{\boldsymbol{\lambda}}\|)$, for all $\boldsymbol{\lambda}, \tilde{\boldsymbol{\lambda}} \in \Lambda$.

These two assumptions are quite weak and can be satisfied by popular continuous distributions, such as the exponential family.

We now provide an end-to-end theorem that guarantees the performance of the PAC-Bayes training based on Equation (5) for general priors and posteriors.

**Theorem 4.15** (PAC-Bayes bound for unbounded losses and **trainable** priors). *Assume the loss $\ell(\mathbf{h}, z_i)$ as a random variable parametrized by model $\mathbf{h}$ satisfies Definition 4.7. Let $\mathfrak{Q}$ be a family of posterior distribution $\mathfrak{Q} = \{Q_{\mathbf{w}}, \mathbf{w} \in \mathbb{R}^d\}$ , let $\mathfrak{P} = \{P_{\boldsymbol{\lambda}}, \boldsymbol{\lambda} \in \Lambda \subseteq \mathbb{R}^k\}$ be a family of prior distributions parameterized by $\boldsymbol{\lambda}$. Let $n(\varepsilon) := \mathcal{N}(\Lambda, \|\cdot\|, \varepsilon)$ be the covering number of the set of the prior parameters. Under Assumption 4.13 and Assumption 4.14, the following inequality holds for the minimizer $(\hat{\mathbf{w}}, \hat{\gamma}, \hat{\boldsymbol{\lambda}})$ of Equation (P) and any $\delta, \varepsilon > 0$ with probability as least $1 - \epsilon$ with $\epsilon := (n(\varepsilon) + \frac{\gamma_2 - \gamma_1}{\varepsilon})\delta$:*

$$\mathbb{E}_{\mathbf{h} \sim \mathcal{Q}_{\hat{\mathbf{w}}}} \ell(\mathbf{h}; \mathcal{D}) \leq L_{PAC}(\hat{\mathbf{w}}, \hat{\gamma}, \hat{\boldsymbol{\lambda}}, \delta) + \eta, \tag{6}$$

*where $\eta = B\varepsilon + C \cdot (\eta_1(\varepsilon) + \eta_2(\varepsilon)) + \frac{\log(n(\varepsilon) + \frac{\gamma_2 - \gamma_1}{2\varepsilon})}{\gamma_1 m}$, and $C$ and $B$ are constants depending on $\gamma_1, \gamma_2$, $m$ and the upper bounds of the parameters in the prior and posterior.*

The proof is available in Appendix A.2.

The theorem provides a generalization bound on the model learned as the minimizer of Equation (P) with data-dependent priors. This bound contains the PAC-Bayes loss $L_{PAC}$ along with an additional correction term $\hat{\eta}$, which is notably absent in the traditional PAC-Bayes bound with fixed priors. Given that $(\hat{\mathbf{w}}, \hat{\gamma}, \hat{\boldsymbol{\lambda}})$ minimizes $L_{PAC}(\cdot, \cdot, \cdot, \delta)$, evaluating $L_{PAC}$ at its own minimizer ensures that the first term is small. If the correction term is also small, then the test error remains low. In the next section, we will delve deeper into the condition for this term to be small. Intuitively, selecting a small $\varepsilon$ helps to maintain low values for the first three terms in $\eta$. Although a smaller $\varepsilon$ increases the $n(\varepsilon)$ in the last term, this increase is moderated because it is inside the logarithm and divided by the size of the dataset.

### 4.4.2 Restricting to the Gaussian Families

For the $L_{PAC}$ objective to have a closed-form formula, we employ the Gaussian distribution family. For ease of illustration, we introduce a new notation for the parametrization. Consider a model denoted as $f_{\boldsymbol{\theta}}$, where $f$ represents the model architecture (e.g., a VGG net), and $\boldsymbol{\theta}$ is the weight. In this context, $f_{\boldsymbol{\theta}}$ aligns with the $\mathbf{h}$ discussed in earlier sections. Moving forward, we will use $f_{\boldsymbol{\theta}}$ to refer to the model instead of $\mathbf{h}$.

We define the posterior distribution of the weights as a Gaussian distribution centered around the trainable weight $\boldsymbol{\mu}$, with trainable variance $\boldsymbol{\sigma}$., i.e., the posterior weight distribution is $\mathcal{N}(\boldsymbol{\mu}, \text{diag}(\boldsymbol{\sigma}))$, denoted by $\mathcal{Q}_{\boldsymbol{\mu}, \boldsymbol{\sigma}}$[1]. The assumption of a diagonal covariance matrix implies the independence of the weights. We consider two types of priors, both centered around the initial weight $\boldsymbol{\mu}_0$ of the model (as suggested by Dziugaite & Roy (2017)), but with different settings on the variance.

**Scalar prior**: we use a universal scalar to encode the variance of all the weights in the prior, i.e., the weight distribution of $\mathcal{P}_{\lambda}$ is $\mathcal{N}(\boldsymbol{\mu}_0, \lambda I_d)$, where $\lambda$ is a scalar. With this prior, the KL divergence $\text{KL}(\mathcal{Q}_{\boldsymbol{\mu}, \boldsymbol{\sigma}} || \mathcal{P}_{\boldsymbol{\mu}_0, \lambda})$ in Equation (P) is:

$$\frac{1}{2}\left[-\mathbf{1}_d^\top \log(\boldsymbol{\sigma}) + d(\log(\lambda) - 1) + \frac{(\|\boldsymbol{\sigma}\|_1 + \|\boldsymbol{\mu} - \boldsymbol{\mu}_0\|_2^2)}{\lambda}\right]. \tag{7}$$

**Layerwise prior**: weights in the $i$th layer share a common variance $\boldsymbol{\lambda}_i$, but different layers could have different variances. By setting $\boldsymbol{\lambda} = (\boldsymbol{\lambda}_1, ...., \boldsymbol{\lambda}_k)$ as the vector containing all the layerwise variances of a $k$-layer

---

[1]A clarification of the notation: we have defined several model-related notations $\mathbf{h}$, $\mathbf{w}$, $\boldsymbol{\theta}$, $\boldsymbol{\mu}$, and $\boldsymbol{\sigma}$. To clarify, their relations are $\mathbf{h} = f_{\boldsymbol{\theta}}$, $\boldsymbol{\theta} \sim \mathcal{Q}_{\boldsymbol{\mu}, \boldsymbol{\sigma}} \equiv \mathcal{Q}_{\mathbf{w}}$, and $\mathbf{w} = [\boldsymbol{\mu}, \boldsymbol{\sigma}]$

neural network, the weight distribution of prior $\mathcal{P}_{\boldsymbol{\lambda}}$ is $\mathcal{N}(\boldsymbol{\mu}_0, \text{BlockDiag}(\boldsymbol{\lambda}))$, where $\text{BlockDiag}(\boldsymbol{\lambda})$ is obtained by diagonally stacking all $\boldsymbol{\lambda}_i I_{d_i}$ into a $d \times d$ matrix, where $d_i$ is the number of weights of the $i$th layer. The KL divergence for layerwise prior is in Appendix A.3. For shallow networks, it is enough to use the scalar prior; for deep neural networks and neural networks constructed from different types of layers, using the layerwise prior is more sensible.

By plugging in the closed-form Equation (7) for $\text{KL}(\mathcal{Q}_{\boldsymbol{\mu},\boldsymbol{\sigma}}||\mathcal{P}_{\boldsymbol{\mu}_0,\boldsymbol{\lambda}})$ into the PAC-Bayes bound in Theorem 4.15, we have the following corollary that justifies the usage of PAC-Bayes bound on large neural networks with the trainable prior.

**Corollary 4.16** (Validity of trainable Gaussian priors)**.** *Suppose the posterior and prior are Gaussian distributions as defined above. Assume all parameters for the prior and posterior are bounded, i.e., we restrict the model parameter $\boldsymbol{\mu}$, the posterior variance $\boldsymbol{\sigma}$ and the prior variance $\boldsymbol{\lambda}$, all to be searched over bounded sets, $\Theta := \{\boldsymbol{\mu} \in \mathbb{R}^d : \|\boldsymbol{\mu}\|_2 \leq \sqrt{d}M\}$, $\Sigma := \{\boldsymbol{\sigma} \in \mathbb{R}^d_+ : \|\boldsymbol{\sigma}\|_1 \leq dT\}$, $\Lambda =: \{\boldsymbol{\lambda} \in [e^{-a}, e^b]^k\}$, respectively, with fixed $M, T, a, b > 0$. Then,*

- *Assumption 4.13 holds with $\eta_1(x) = L_1 x$, where $L_1 = \frac{1}{2}\max\{d, e^a(2\sqrt{d}M + dT)\}$*
- *Assumption 4.14 holds with $\eta_2(x) = L_2 x$, where $L_2 = \frac{1}{\gamma_1^2}\left(2dM^2 e^{2a} + \frac{d(a+b)}{2}\right)$*
- *With high probability, the PAC-Bayes bound for the minimizer of Equation (P) has the form*

$$\mathbb{E}_{\boldsymbol{\theta} \sim \mathcal{Q}_{\hat{\boldsymbol{\mu}},\hat{\boldsymbol{\sigma}}}} \ell(f_{\boldsymbol{\theta}}; \mathcal{D}) \leq L_{PAC}([\hat{\boldsymbol{\mu}}, \hat{\boldsymbol{\sigma}}], \hat{\gamma}, \hat{\boldsymbol{\lambda}}, \delta) + \eta,$$

*where $\eta = \frac{k}{\gamma_1 m}\left(1 + \log\frac{2(CL+B)\Delta\gamma_1 m}{k}\right)$, $L = L_1 + L_2$, $\Delta := \max\{b+a, 2(\gamma_2 - \gamma_1)\}$, $C = \frac{1}{\gamma_1 m} + \gamma_2$ $B$ is a constant depending on $\gamma_1, \delta, M, d, T, a, b, m^2$.*

In the bound, the term $L_{PAC}([\hat{\boldsymbol{\mu}}, \hat{\boldsymbol{\sigma}}], \hat{\gamma}, \hat{\boldsymbol{\lambda}}, \delta)$ is inherently minimized as it evaluates the function $L_{PAC}(\cdot, \cdot, \cdot, \delta)$ at its own minimizer. The overall bound remains low if the correction term $\eta$ can be deemed insignificant. The logarithm term in the definition of $\eta$ grows very mildly with the dimension in general, so we can treat it (almost) as a constant. Thus, $\eta \sim \frac{k}{\gamma_1 m}$, from which we see that 1). $\eta$ (and therefore the bound) would be small if prior's degree of freedom $k$ is substantially less than the dataset size $m$; 2). This bound still achieves the asymptotic rate of $O(m^{-1/2})$ after optimizing over $\gamma_1$. We note that even if the corollary assumes that the parameters (i.e., mean and variance) of the Gaussian distribution are bounded, the random variable itself is still unbounded, so the loss is still unbounded. The proof and more discussions can be found in Appendix A.4.

## 4.5 Training algorithm

**Estimating** $K_{\min}(\boldsymbol{\lambda})$**:** In practice, the function $K_{\min}(\boldsymbol{\lambda})$ must be estimated first. Since we showed in Corollary 4.16 and Remark 4.6 that $K_{\min}(\boldsymbol{\lambda})$ is Lipschtiz continuous and bounded, we can approximate it using piecewise-linear functions. More explicitly, we use Equation (3) and Monte Carlo Sampling to estimate $K_{\min}$ on some discrete grid of $\boldsymbol{\lambda}$, then we interpolate $K_{\min}(\boldsymbol{\lambda})$ using piecewise linear functions. More details are in Appendix B.1.

Notably, since for each fixed $\boldsymbol{\lambda} \in \Lambda$, the prior is independent of the training data, this procedure of estimating $K_{\min}(\boldsymbol{\lambda})$ can be carried out before training. Recall that after Remark 4.8, we discussed two ways to estimate $K_{\min}$, using Equation (3) or using its theoretical upper bound from Lemma 4.5. Figure 2 illustrates the advantage of the former approach. Besides having a much smaller numerical value, estimating $K_{\min}$ directly from Equation (3) also involves fewer constraints compared to using the upper bound, which requires a non-negative loss with a bounded second-order moment.

**Two-stage PAC-Bayes training:** Algorithm 1 outlines the proposed PAC-Bayes training algorithm with scalar prior. The version that uses the layerwise prior is detailed in Appendix B.2. Algorithm 1 contains two stages. Stage 1 performs pure PAC-Bayes training, and Stage 2 is an optional Bayesian refinement stage that is only activated when the first stage does not sufficiently reduce the training loss. For Stage 1, although there are several input parameters to be specified, one can use the same choice of values across very different

---

[2]See Appendix A.4 for the explicit form of $B$.

---

**Algorithm 1** PAC-Bayes training (scalar prior)

---

**Input:** initial weight $\boldsymbol{\mu}_0 \in \mathbb{R}^d$, $T_1 = 500$, $\lambda_1 = e^{-12}$, $\lambda_2 = e^2$, $\gamma_1 = 0.5, \gamma_2 = 10$. // $T_1, \lambda_1, \lambda_2, \gamma_1, \gamma_2$ *are fixed in all experiments of Sec.5.*
**Output:** trained weight $\hat{\boldsymbol{\mu}}$, posterior noise level $\hat{\boldsymbol{\sigma}}$
$\boldsymbol{\mu} \leftarrow \boldsymbol{\mu}_0$, $\mathbf{v} \leftarrow \mathbf{1_d} \cdot \log(\frac{1}{d}\|\boldsymbol{\mu}_0\|_1)$, $b \leftarrow \log(\frac{1}{d}\|\boldsymbol{\mu}_0\|_1)$
Obtain $\hat{K}(\lambda)$ with $\Lambda = [\lambda_1, \lambda_2]$ using Equation (26) (Appendix Algorithm 2)
*/*Stage 1*/*
**for** epoch $= 1 : T_1$ **do**
    **for** sampling one batch $s$ from $\mathcal{S}$ **do**
        *//Ensure non-negative variances*
        $\lambda \leftarrow \exp(b)$, $\boldsymbol{\sigma} \leftarrow \exp(\mathbf{v})$
        $\mathcal{P}_\lambda \leftarrow \mathcal{N}(\boldsymbol{\mu}_0; \lambda I_d)$, $\mathcal{Q}_{\boldsymbol{\mu},\boldsymbol{\sigma}} \leftarrow \boldsymbol{\mu} + \mathcal{N}(\mathbf{0}; \mathrm{diag}(\boldsymbol{\sigma}))$
        *//Get the stochastic version of $\mathbb{E}_{\tilde{\boldsymbol{\theta}} \sim \mathcal{Q}_{\boldsymbol{\mu},\sigma}} \ell(f_{\tilde{\boldsymbol{\theta}}}; \mathcal{S})$*
        Draw one $\tilde{\boldsymbol{\theta}} \sim \mathcal{Q}_{\boldsymbol{\mu},\boldsymbol{\sigma}}$ and evaluate $\ell(f_{\tilde{\boldsymbol{\theta}}}; \mathcal{S})$
        Compute the KL divergence as Equation (7)
        Compute $\gamma$ as Equation (8)
        Compute the loss function $\mathcal{L}$ as $L_{PAC}$ in Equation (P)
        *//Update all parameters*
        $b \leftarrow b + \eta\frac{\partial\mathcal{L}}{\partial b}$, $\mathbf{v} \leftarrow \mathbf{v} + \eta\frac{\partial\mathcal{L}}{\partial\mathbf{v}}$, $\boldsymbol{\mu} \leftarrow \boldsymbol{\mu} + \eta\frac{\partial\mathcal{L}}{\partial\boldsymbol{\mu}}$
    **end for**
**end for**
*//Fix the noise level from now on*
$\hat{\boldsymbol{\sigma}} \leftarrow \exp(\mathbf{v})$
*/*Stage 2*/* // *Run this stage if the training has not fully converged during Stage 1*
**while** not converge **do**
    **for** sampling one batch $s$ from $\mathcal{S}$ **do**
        *//Noise injection*
        Draw one $\tilde{\boldsymbol{\theta}} \sim \mathcal{Q}_{\hat{\boldsymbol{\mu}},\hat{\boldsymbol{\sigma}}}$ and evaluate $\ell(f_{\tilde{\boldsymbol{\theta}}}; \mathcal{S})$ as $\tilde{\mathcal{L}}$,
        *//Update model parameters*
        $\boldsymbol{\mu} \leftarrow \boldsymbol{\mu} + \eta\frac{\partial\tilde{\mathcal{L}}}{\partial\boldsymbol{\mu}}$
    **end for**
**end while**
$\hat{\boldsymbol{\mu}} \leftarrow \boldsymbol{\mu}$

---

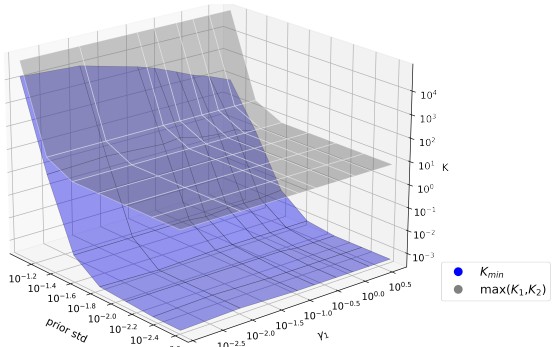

Figure 2: Comparison of the upper bound $\max\{K_1, K_2\}$ for $K_{\min}$, as derived in Lemma 4.5 with the data-driven estimate of $K_{\min}$ obtained via Equation (3) on CNN9 using the CIFAR10 dataset with the prior parameterized as a Gaussian distribution centered at the Kaiming initialization.

network architectures and datasets with minor modifications. Please see Appendix C.1 for more discussions. When everything else in the PAC-Bayes loss is fixed, $\gamma \in [\gamma_1, \gamma_2]$ has a closed-form solution,

$$\gamma^* = \min \left\{ \max \left\{ \gamma_1, \sqrt{\frac{\log \frac{1}{\delta} + \mathrm{KL}(\mathcal{Q}_{\boldsymbol{\mu},\boldsymbol{\sigma}} || \mathcal{P}_{\boldsymbol{\mu}_0, \boldsymbol{\lambda}})}{m K_{\min}}} \right\}, \gamma_2 \right\} \tag{8}$$

Therefore, we only need to perform gradient updates on the other three variables, $\boldsymbol{\mu}, \boldsymbol{\sigma}, \boldsymbol{\lambda}$.

**The second stage of training:** Gastpar et al. (2023); Nagarajan & Kolter (2019) showed that achieving high accuracy on certain distributions precludes the possibility of getting a tight generalization bound in overparameterized settings. This implies that it is less possible to use reasonable generalization bound to fully train one overparameterized model on a particular dataset. It is also observed in our PAC-Bayes training experiments that, oftentimes, minimizing the PAC-Bayes bound only (Stage 1) cannot make the training accuracy reach 100%. If this happens[3], we add a second stage to further increase the training accuracy. Specifically, in Stage 2, we continue to update the model by minimizing only $\mathbb{E}_{\boldsymbol{\theta} \sim \mathcal{Q}_{\boldsymbol{\mu},\hat{\boldsymbol{\sigma}}}} \ell(f_{\boldsymbol{\theta}}; \mathcal{S})$ over $\boldsymbol{\mu}$, and keep all other variables (i.e., $\boldsymbol{\lambda}$, $\boldsymbol{\sigma}$) fixed to the solution found by Stage 1. This is essentially a stochastic gradient descent with noise injection, the level of which has been learned from Stage 1. The two-stage training is similar to the idea of the learning-rate scheduler (LRS). In LRS, the initial large learning rate introduces an implicit bias that guides the solution path towards a flat region (Cohen et al., 2021; Barrett & Dherin, 2020), and the later lower learning rate ensures the convergence to a local minimizer in this region. Without the large learning rate stage, it cannot reach the flat region; without the small learning rate stage, it cannot converge to a local minimizer. For the two-stage PAC-Bayes training, Stage 1 (PAC-Bayes stage) guides the solution to flat regions by minimizing the generalization bound, and Stage 2 is necessary for an actual convergence to a local minimizer.

**Regularizations in the PAC-Bayes training:** By plugging the KL divergence Equation (7) into P, we can see that in the case of Gaussian priors and posteriors, the PAC-Bayes loss is nothing but the original training loss augmented by a noise injection and a weight decay, except that strength of both of them are automatically learned during training. More discussions are available in Appendix B.3.

**Prediction:** After training, we use the mean of the posterior as the trained model and perform deterministic prediction on the test dataset. In Appendix B.4, we provide some mathematical intuition of why the deterministic predictor is expected to perform even better than the Bayesian predictor.

## 5 Experiments

In this section, we demonstrate the efficacy of the proposed PAC-Bays training algorithm through extensive numerical experiments. Specifically, we conduct comparisons between our algorithm and existing PAC-Bayes training algorithms, as well as conventional training algorithms based on Empirical Risk Minimization (ERM). Our approach yields competitive test accuracy in all settings and exhibits a high degree of robustness w.r.t. the choice of hyperparameters.

**Comparison with different PAC-Bayes bounds and existing PAC-Bayes training algorithms:** We compared our PAC-Bayes training algorithm using the layerwise prior with baselines in Pérez-Ortiz et al. (2021): *quad* (Rivasplata et al., 2019), *lambda* (Thiemann et al., 2017), *classic* (McAllester, 1999), and *bbb* (Blundell et al., 2015) in the context of deep convolutional neural networks. The baseline PAC-Bayes algorithms contain a variety of crucial hyperparameters, including variance of the prior (1e-2 to 5e-6), learning rate (1e-3 to 1e-2), momentum (0.95, 0.99), dropout rate (0 to 0.3) in the training of the prior, and the KL trade-off coefficient (1e-5 to 0.1) for *bbb*. These hyperparameters were chosen by grid search. The batch size is 250 for all methods. Our findings, as detailed in Table 1, show that our algorithm outperforms the other PAC-Bayes methods regarding test accuracy. It is important to note that all four baselines employed the PAC-Bayes bound for bounded loss. Therefore, they need to convert unbounded loss into bounded loss for training purposes. Various conversion methods were evaluated by Pérez-Ortiz et al. (2021), and the most effective one was selected for producing the results presented.

---

[3]Note that there are cases when the second stage is not necessary, including but not limited to 1) the network is shallow 2) the dataset is simple 3) a good prior is chosen. In these cases, the training accuracy can already reach 100% in Stage 1.

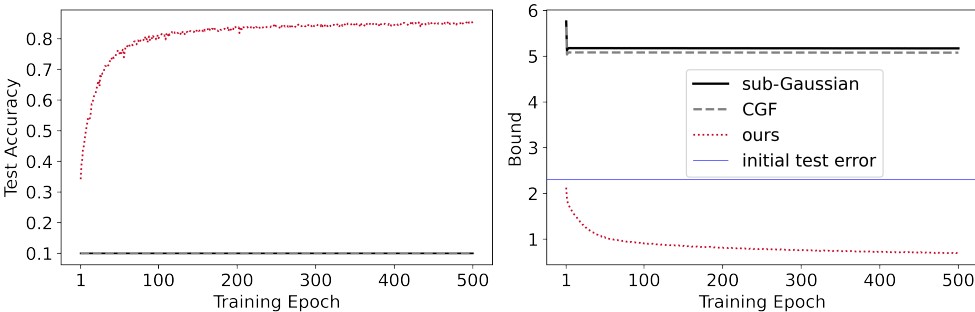

Figure 3: Minimizing PAC-Bayes bounds based on *sub-Gaussian*, *CGF* and our proposed bound on CIFAR10 using CNN9. The test error of a randomly initialized model is shown as *initial test error*. Minimizing our bound (*ours*) achieves a better test accuracy compared with optimizing the other two (*sub-Gaussian* and *CGF*).

Table 1: Test accuracy of convolution neural networks on CIFAR10. The test accuracy of baselines for bounded loss is from Table 5 of Pérez-Ortiz et al. (2021), calculated as 1-the zero-one error of the deterministic predictor. *subG* represents the sub-Gaussian bound. Our proposed PAC-Bayes training with a layerwise prior (*layer*) achieves the best test accuracy across all models.

|       | bounded | | | | unbounded | | |
|-------|------|--------|---------|-------|-------|-------|-------|
|       | quad | lambda | classic | bbb | subG | CGF | layer |
| CNN9  | 78.63 | 79.39 | 78.33 | 83.49 | 78.53 | 78.35 | **85.46** |
| CNN13 | 84.47 | 84.48 | 84.22 | 85.41 | 84.30 | 84.42 | **88.31** |
| CNN15 | 85.31 | 85.51 | 85.20 | 85.95 | 84.98 | 85.13 | **87.55** |

To demonstrate the necessity of our newly proposed PAC-Bayes bound for unbounded loss, we compared this new bound with two existing PAC-Bayes bounds for unbounded loss. One is based on the *sub-Gaussian* assumption (Theorem 5.1 of Alquier (2021)), while the other (Theorem 9 of Rodríguez-Gálvez et al. (2023)) assumes the loss function is a bounded cumulant generating function (*CGF*). It is important to note that, as of now, no training algorithms specifically leverage these PAC-Bayes bounds for unbounded loss. Therefore, for a fair comparison, we conducted an experiment by replacing our PAC-Bayes bound with the other two bounds and using the same two-stage training algorithm with the trainable layerwise prior.

We also visualized the test accuracy when minimizing different PAC-Bayes bounds for unbounded loss in Stage 1. As shown in Figure 3, minimizing our PAC-Bayes bound can achieve better generalization performance. As discussed in Remark 4.12, the $K$ terms in the two baseline bounds for unbounded loss are much larger compared to ours. This results in a smaller $\gamma$, which increases the coefficient of the KL divergence term and forces the posterior to remain close to the prior rather than fitting the data effectively. The details of the two baseline bounds are in Appendix C.2.

**Comparison with ERM optimized by SGD/Adam with various regularizations:** We tested our PAC-Bayes training on CIFAR10 and CIFAR100 datasets with *no data augmentation*[4] on various popular deep neural networks, VGG13, VGG19 (Simonyan & Zisserman, 2014), ResNet18, ResNet34 (He et al., 2016), and Dense121 (Huang et al., 2017) by comparing its performance with conventional empirical risk minimization by SGD/Adam enhanced by various regularizations (which we call baselines). The training of baselines involves a grid search for the best hyperparameters, including momentum for SGD (0.3 to 0.9), learning rate (1e-3 to 0.2), weight decay (1e-4 to 1e-2), and noise injection (5e-4 to 1e-2). The batch size was set to be 128. We reported the highest test accuracy obtained from this search as the baseline results. For all convolutional neural networks, our method employed Adam with a fixed learning rate of 1e-4.

---

[4]Result with data augmentation can be found in Appendix C.3

Table 2: Test accuracy of CNNs on C10 (CIFAR10) and C100 (CIFAR100) with batch size 128. Our PAC-Bayes training with scalar and layerwise prior are labeled *scalar* and *layer*. The best and second-best test accuracies are **highlighted** and underlined. Our PAC-Bayes training can approximately match the best performance of the baseline.

| | VGG13 | | VGG19 | | ResNet18 | | ResNet34 | | Dense121 | |
|---|---|---|---|---|---|---|---|---|---|---|
| | C10 | C100 | C10 | C100 | C10 | C100 | C10 | C100 | C10 | C100 |
| SGD | **90.2** | 66.9 | 90.2 | **64.5** | **89.9** | 64.0 | 90.0 | **70.3** | **91.8** | **74.0** |
| Adam | 88.5 | 63.7 | 89.0 | 58.8 | 87.5 | 61.6 | 87.9 | 59.5 | 91.2 | 70.0 |
| AdamW | 88.4 | 61.8 | 89.0 | 62.3 | 87.9 | 61.4 | 88.3 | 59.9 | 91.5 | 70.1 |
| scalar | 88.7 | **67.2** | 89.2 | 61.3 | 88.0 | 68.8 | 89.6 | 69.5 | 91.2 | 71.4 |
| layer | 89.7 | 67.1 | **90.5** | 62.3 | 89.3 | **68.9** | **90.9** | 69.9 | 91.5 | 72.2 |

Table 3: Test accuracy of GNNs trained with AdamW versus our proposed method with scalar prior *scalar*. The best test accuracies are **highlighted**. The performance of our training can almost match the best results of the baseline obtained after carefully tuning hyperparameters.

| | | CoraML | Citeseer | PubMed | Cora | DBLP |
|---|---|---|---|---|---|---|
| GCN | AdamW | 85.7±0.7 | **90.3**±0.4 | **85.0**±0.6 | 60.7±0.7 | **80.6**±1.4 |
| | scalar | **86.1**±0.7 | 90.0±0.4 | 84.9±0.8 | **62.0**±0.4 | 80.5±0.6 |
| GAT | AdamW | 85.7±1.0 | **90.8**±0.3 | 84.0±0.4 | **63.5**±0.4 | **81.8**±0.6 |
| | scalar | **85.9**±0.8 | 90.6±0.5 | **84.4**±0.5 | 60.9±0.6 | 81.0±0.5 |
| SAGE | AdamW | 85.7±0.5 | **90.5**±0.5 | 83.5±0.4 | 60.6±0.5 | **80.7**±0.6 |
| | scalar | **86.5**±0.5 | 90.0±0.5 | **84.4**±0.6 | **61.2**±0.2 | 79.9±0.5 |
| APPNP | AdamW | 86.6±0.7 | **91.0**±0.4 | 85.1±0.5 | 62.5±0.4 | 80.6±2.8 |
| | scalar | **87.1**±0.6 | 90.4±0.5 | **85.7**±0.4 | **63.5**±0.4 | **81.8**±0.5 |

Since the CIFAR10 and CIFAR100 datasets do not have a published validation dataset, **we used the test dataset to find the best hyperparameters of baselines during the grid search, which might lead to a slightly inflated performance for baselines.** Nevertheless, as presented in Table 2, the test accuracy of our method is still competitive. Please refer to Appendix C.4 for more details.

**Evaluation on graph neural networks:** To demonstrate the broad applicability of the proposed PAC-Bayes training algorithm to different network architectures, we evaluated it on graph neural networks (GNNs). Unlike CNNs, optimal GNN performance has been reported using the AdamW optimizer for ERM and enabling dropout. To ensure the best baseline results, we conducted a hyperparameter search over learning rate (1e-3 to 1e-2), weight decay (0 to 1e-2), noise injection (0 to 1e-2), and dropout (0 to 0.8) and reported the highest test accuracy as the baseline result. For our method, we used Adam and fixed the learning rate to be 1e-2 for all graph neural networks. We follow the convention for graph datasets by randomly assigning 20 nodes per class for training, 500 for validation, and the remaining for testing.

We tested four architectures GCN (Kipf & Welling, 2016), GAT (Veličković et al., 2017), SAGE (Hamilton et al., 2017), and APPNP (Gasteiger et al., 2018) on 5 benchmark datasets CoraML, Citeseer, PubMed, Cora and DBLP (Bojchevski & Günnemann, 2017). Since there are only two convolution layers for GNNs, applying our algorithm with the scalar prior is sensible. For our PAC-Bayes training, we retained the dropout layer in the GAT as is, since it differs from the conventional dropout and essentially drops the edges of the input graph. Other architectures do not have this type of dropout; hence, our PAC-Bayes training for these architectures does not include dropout.

Table 4: The test accuracy for CNNs on CIFAR10 (C10) and CIFAR100 (C100) using a batch size of 2048. Values in (·) indicate how much the results differ from using a batch size (128). Our PAC-Bayes training with scalar and layerwise prior are labeled as *scalar* and *layer*. The most robust results w.r.t. the increase of batch size are **highlighted**, indicating the elevated robustness of our method compared to the baseline regarding batch sizes.

| | VGG13 | | ResNet18 | |
|---|---|---|---|---|
| | C10 | C100 | C10 | C100 |
| SGD | 87.7 (-2.5) | 60.1 (-6.8) | 85.4 (-4.5) | 61.5 (-2.6) |
| Adam | 90.7 (+2.2) | 66.2 (+2.5) | 87.7 (+0.2) | 65.4 (+3.8) |
| AdamW | 87.2 (-1.1) | 61.0 (-0.8) | 84.9 (-2.9) | 58.9 (-2.5) |
| scalar | 88.9 (**+0.2**) | 66.0 (-1.2) | 88.9 (+0.9) | 68.7 (**-0.1**) |
| layer | 89.4 (-0.3) | 67.1 (**0.0**) | 89.2 (**-0.1**) | 69.3 (+0.3) |

Table 5: Test accuracy of ResNet18 and VGG13 trained with different learning rates on CIFAR10. The best test accuracies are **highlighted**. Our method is more robust to learning rate variations.

| Model | Method | 3e-5 | 5e-5 | 1e-4 | 2e-4 | 3e-4 | 5e-4 | 1e-3 |
|---|---|---|---|---|---|---|---|---|
| ResNet18 | layer | **88.4** | **88.8** | **89.3** | **88.6** | **88.3** | **89.2** | 87.3 |
| | Adam | 66.6 | 73.9 | 81.2 | 85.3 | 86.4 | 87.0 | **87.5** |
| VGG13 | layer | **88.6** | **88.9** | **89.7** | **89.6** | **89.6** | **89.5** | **88.7** |
| | Adam | 84.3 | 84.8 | 85.8 | 87.4 | 87.9 | 88.3 | 88.5 |

Table 3 demonstrates that the performance of our algorithm closely approximates the best outcome of the baseline. Appendix C.5 provides additional details and more results. Extra analysis on few-shot text classification with transformers is in Appendix C.6.

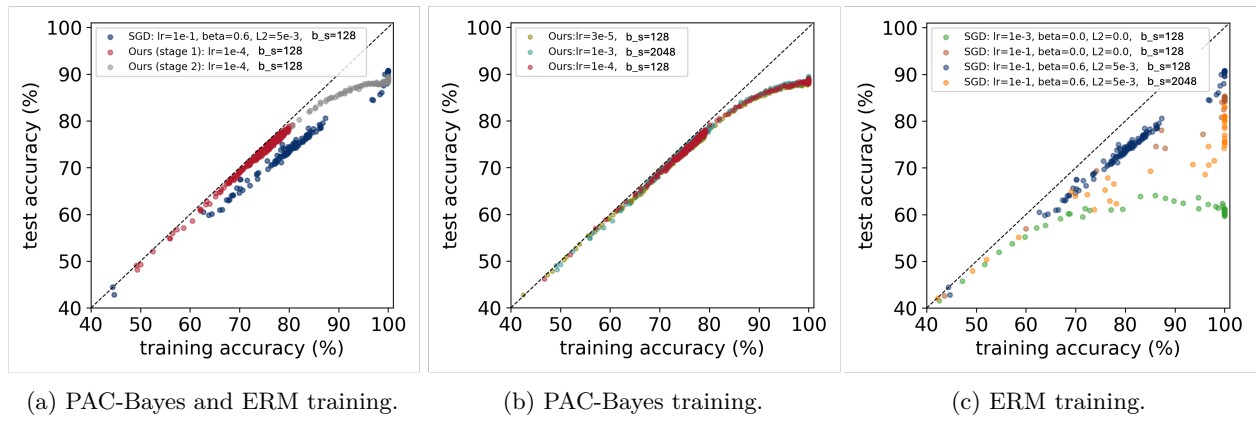

(a) PAC-Bayes and ERM training.  (b) PAC-Bayes training.  (c) ERM training.

Figure 4: Generalization gap (the difference between the training and the testing accuracy) in PAC-Bayes training versus ERM training using ResNet18 on the CIFAR10 dataset. Each point represents an intermediate model during training, plotted according to its test accuracy versus training accuracy. The line $y = x$ indicates the optimal, zero generalization gap. PAC-Bayes training has a smaller generalization gap throughout the training process (Fig. 4a) and remains stable despite changes in hyperparameters (Fig. 4b). In constant, ERM training (Fig. 4c) is very unstable to hyper-parameter changes. When comparing ERM with our method in Fig. 4a, we picked the best ERM result (the blue one) in Fig. 4c that achieved the best final test accuracy. The discontinuity it has around the testing accuracy of 87%, is due to the activation of the learning rate scheduler.

**Evaluation on the sensitivity of hyperparameters**: In previous experiments, we selected specific batch sizes and learning rates as the only two tunable hyperparameters of our algorithm, with all other parameters remaining constant across all experiments. We further demonstrate that batch size and learning rate variations do not significantly impact our final performance. This suggests a general robustness of our method to hyperparameters, reducing the necessity for extensive tuning. More specifically, with a fixed learning rate 5e-4 in our method, Table 4 shows that changing the batch size from 128 to a very large one, 2048, for VGG13 and ResNet18 does not significantly affect the performance of the PAC-Bayes training compared to ERM with extensive tuning as before. Also, as shown by Table 5, our algorithm is more robust to learning rate changes than ERM, which utilizes the optimal weight decay and noise injection settings from Table 2.

We also examine the change in the generalization gap across the training process Figure 4. Generalization gap is defined to be the difference between the training and testing accuracy. Algorithms with better generalization ability should yield a smaller generalization gap. The smallest possible gap is 0, which corresponds to the line $y = x$ in Figure 4. We observe in Figure 4a that our PAC-Bayes training, when compared to the ERM with the optimal hyperparameter setting, has a smaller generalization gap (i.e., closer to the line $y = x$) over the course of training, although the final test accuracies are similar. The generalization gap in Stage 1 is extremely small, confirming the effectiveness of using the PAC-Bayes bound for achieving good generalization. In addition, as we vary the choice of hyperparameters, PAC-Bayes training is much more stable Figure 4b than ERM Figure 4c, indicating less need for hyperparameter tuning. Furthermore, Fig. 4 suggests future directions for improving the PAC-Bayes training. Since Stage 1 yields the best generalization gap, we should focus on developing numerically tighter PAC-Bayes bounds to prolong Stage 1. Alternatively, we can aim to improve the heuristic algorithm in Stage 2 to minimize the increase in the generalization gap during this stage.

Please refer to Appendix C.7 for extra experimental studies.

## 6 Conclusion and Discussion

In this paper, we demonstrated the great practical potential of PAC-Bayes training by proposing a numerically tighter PAC-Bayes bound and applying it to train deep neural networks. The proposed framework significantly improved the performance of PAC-Bayes training, making it nearly match the best results of ERM. We hope this result inspires researchers in the field to further explore the practical implications of PAC-Bayes theory and make these bounds more useful in practice.

## Acknowledgments

We thank Andrew Gordon Wilson for insightful and valuable discussions. A.G. was supported by NSF grant CCF-2212065. R.W. was partially supported by NSF grant CCF-2212065 and BCS-2215155.

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

# Appendix

# A Proofs

## A.1 Proofs of Theorem 4.9

**Theorem A.1.** *Given a prior $\mathcal{P}_{\boldsymbol{\lambda}}$ parametrized by $\boldsymbol{\lambda} \in \Lambda$ over the hypothesis set $\mathcal{H}$. Fix $\boldsymbol{\lambda} \in \Lambda$, $\delta \in (0,1)$ and $\gamma \in [\gamma_1, \gamma_2]$. For any choice of i.i.d $m$-sized training dataset $\mathcal{S}$ according to $\mathcal{D}$, and all posterior distributions $\mathcal{Q}$ over $\mathcal{H}$, we have*

$$\mathbb{E}_{\mathbf{h} \sim \mathcal{Q}} \ell(\mathbf{h}; \mathcal{D}) \leq \mathbb{E}_{\mathbf{h} \sim \mathcal{Q}} \ell(\mathbf{h}; \mathcal{S}) + \frac{1}{\gamma m}(\log \frac{1}{\delta} + \mathrm{KL}(\mathcal{Q}||\mathcal{P}_{\boldsymbol{\lambda}})) + \gamma K(\boldsymbol{\lambda}) \tag{9}$$

*holds with probability at least $1 - \delta$ when $\ell(\mathbf{h}, \cdot)$ satisfies Definition 4.7 with bound $K(\boldsymbol{\lambda})$.*

**Proof.** Firstly, in the bounded interval $\gamma \in [\gamma_1, \gamma_2]$, we bound the difference of the expected loss over the posterior distribution evaluated on the training dataset $\mathcal{S}$ and $\mathcal{D}$ with the KL divergence between the posterior distribution $\mathcal{Q}$ and prior distribution $\mathcal{P}_{\boldsymbol{\lambda}}$ evaluated over a hypothesis space $\mathcal{H}$.

For $\gamma \in [\gamma_1, \gamma_2]$,

$$\mathbb{E}_{\mathcal{S} \sim \mathcal{D}}[\exp\left(\gamma m(\mathbb{E}_{\mathbf{h} \sim \mathcal{Q}} \ell(\mathbf{h}; \mathcal{D}) - \mathbb{E}_{\mathbf{h} \sim \mathcal{Q}} \ell(\mathbf{h}; \mathcal{S})) - \mathrm{KL}(\mathcal{Q}||\mathcal{P}_{\boldsymbol{\lambda}}))]$$

$$= \mathbb{E}_{\mathcal{S} \sim \mathcal{D}}[\exp\left(\gamma m(\mathbb{E}_{\mathbf{h} \sim \mathcal{Q}} \ell(\mathbf{h}; \mathcal{D}) - \mathbb{E}_{\mathbf{h} \sim \mathcal{Q}} \ell(\mathbf{h}; \mathcal{S})) - \mathbb{E}_{\mathbf{h} \sim \mathcal{Q}} \log \frac{\mathrm{d}\mathcal{Q}}{\mathrm{d}\mathcal{P}_{\boldsymbol{\lambda}}}(\mathbf{h}))] \tag{10}$$

$$\leq \mathbb{E}_{\mathcal{S} \sim \mathcal{D}} \mathbb{E}_{\mathbf{h} \sim \mathcal{Q}}[\exp\left(\gamma m(\ell(\mathbf{h}; \mathcal{D}) - \ell(\mathbf{h}; \mathcal{S})) - \log \frac{\mathrm{d}\mathcal{Q}}{\mathrm{d}\mathcal{P}_{\boldsymbol{\lambda}}}(\mathbf{h}))] \tag{11}$$

$$= \mathbb{E}_{\mathbf{h} \sim \mathcal{P}_{\boldsymbol{\lambda}}} \mathbb{E}_{\mathcal{S} \sim \mathcal{D}}[\exp(\gamma m(\ell(\mathbf{h}; \mathcal{D}) - \ell(\mathbf{h}; \mathcal{S})))], \tag{12}$$

where $\mathrm{d}\mathcal{Q}/\mathrm{d}\mathcal{P}$ denotes the Radon-Nikodym derivative.

In Equation (10), we use $\mathrm{KL}(\mathcal{Q}||\mathcal{P}_{\boldsymbol{\lambda}}) = \mathbb{E}_{\mathbf{h} \sim \mathcal{Q}} \log \frac{\mathrm{d}\mathcal{Q}}{\mathrm{d}\mathcal{P}_{\boldsymbol{\lambda}}}(\mathbf{h})$. From Equation (10) to Equation (11), Jensen's inequality is used over the convex exponential function. Since this argument holds for any $Q$, we have

$$\sup_{\mathcal{Q} \in \mathbf{Q}} \mathbb{E}_{\mathcal{S} \sim \mathcal{D}}[\exp\left(\gamma m(\mathbb{E}_{\mathbf{h} \sim \mathcal{Q}} \ell(\mathbf{h}; \mathcal{D}) - \mathbb{E}_{\mathbf{h} \sim \mathcal{Q}} \ell(\mathbf{h}; \mathcal{S})) - \mathrm{KL}(\mathcal{Q}||\mathcal{P}_{\boldsymbol{\lambda}}))] \leq \mathbb{E}_{\mathbf{h} \sim \mathcal{P}_{\boldsymbol{\lambda}}} \mathbb{E}_{\mathcal{S} \sim \mathcal{D}}[\exp(\gamma m(\ell(\mathbf{h}; \mathcal{D}) - \ell(\mathbf{h}; \mathcal{S})))] \tag{13}$$

Let $X = \ell(\mathbf{h}; \mathcal{D}) - \ell(\mathbf{h}; \mathcal{S})$, then $X$ is centered with $\mathbb{E}[X] = 0$. Then, by Definition 4.7,

$$\exists K(\boldsymbol{\lambda}), \quad \mathbb{E}_{\mathbf{h} \sim \mathcal{P}_{\boldsymbol{\lambda}}} \mathbb{E}_{\mathcal{S} \sim \mathcal{D}}[\exp\left(\gamma m X\right)] \leq \exp\left(m\gamma^2 K(\boldsymbol{\lambda})\right). \tag{14}$$

Using Markov's inequality, Equation (15) holds with probability at least $1 - \delta$.

$$\exp\left(\gamma m X\right) \leq \frac{\exp\left(m\gamma^2 K(\boldsymbol{\lambda})\right)}{\delta}. \tag{15}$$

Combining Equation (13) and Equation (15), the following inequality holds with probability at least $1 - \delta$.

$$\sup_{\mathcal{Q} \in \mathbb{Q}} \exp\left(\gamma m(\mathbb{E}_{\mathbf{h} \sim \mathcal{Q}} \ell(\mathbf{h}; \mathcal{D}) - \mathbb{E}_{\mathbf{h} \sim \mathcal{Q}} \ell(\mathbf{h}; \mathcal{S})) - \mathrm{KL}(\mathcal{Q}||\mathcal{P}_{\boldsymbol{\lambda}})\right) \leq \frac{\exp\left(m\gamma^2 K(\boldsymbol{\lambda})\right)}{\delta}$$

$$\Rightarrow \gamma m(\mathbb{E}_{\mathbf{h} \sim \mathcal{Q}} \ell(\mathbf{h}; \mathcal{D}) - \mathbb{E}_{\mathbf{h} \sim \mathcal{Q}} \ell(\mathbf{h}; \mathcal{S})) - \mathrm{KL}(\mathcal{Q}||\mathcal{P}_{\boldsymbol{\lambda}}) \leq \log \frac{1}{\delta} + m\gamma^2 K(\boldsymbol{\lambda}), \forall \mathcal{Q}$$

$$\Rightarrow \mathbb{E}_{\mathbf{h} \sim \mathcal{Q}} \ell(\mathbf{h}; \mathcal{D}) \leq \mathbb{E}_{\mathbf{h} \sim \mathcal{Q}} \ell(\mathbf{h}; \mathcal{S}) + \frac{1}{\gamma m}(\log \frac{1}{\delta} + \mathrm{KL}(\mathcal{Q}||\mathcal{P}_{\boldsymbol{\lambda}})) + \gamma K(\boldsymbol{\lambda}), \quad \forall \mathcal{Q}. \tag{16}$$

The bound 16 is exactly the statement of the Theorem.

$\square$

## A.2 Proof of Theorem 4.15

**Theorem A.2.** *Let $n(\varepsilon) := \mathcal{N}(\Lambda, \|\cdot\|, \varepsilon)$ be the covering number of the set of the prior parameters. Under Assumption 4.13 and Assumption 4.14, the following inequality holds for the minimizer $(\hat{\gamma}, \hat{\mathbf{w}}, \hat{\boldsymbol{\lambda}})$ of upper bound in Equation (9) with probability as least $1 - \epsilon$:*

$$\mathbb{E}_{\mathbf{h} \sim \mathcal{Q}_{\hat{\mathbf{w}}}} \ell(\mathbf{h}; \mathcal{D}) \leq \mathbb{E}_{\mathbf{h} \sim \mathcal{Q}_{\hat{\mathbf{w}}}} \ell(\mathbf{h}; \mathcal{S}) + \frac{1}{\hat{\gamma} m} \left[ \log \frac{n(\varepsilon) + \frac{\gamma_2 - \gamma_1}{\varepsilon}}{\epsilon} + \mathrm{KL}(\mathcal{Q}_{\hat{\mathbf{w}}} || \mathcal{P}_{\hat{\boldsymbol{\lambda}}}) \right] + \hat{\gamma} K(\hat{\boldsymbol{\lambda}}) + \eta$$

$$= L_{PAC}(\mathbf{w}, \hat{\gamma}, \hat{\boldsymbol{\lambda}}, \delta) + \eta \tag{17}$$

*holds for any $\epsilon, \varepsilon > 0$, where $\eta = B\varepsilon + C(\eta_1(\varepsilon) + \eta_2(\varepsilon)) + \frac{\log(n(\varepsilon) + \frac{\gamma_2 - \gamma_1}{\varepsilon})}{\gamma_1 m}$, with $C = \frac{1}{\gamma_1 m} + \gamma_2$, and $B := \sup_{\boldsymbol{\lambda} \in \Lambda} \frac{1}{m \gamma_1^2} (\mathrm{KL}(\mathcal{Q}_{\hat{\mathbf{w}}} || \mathcal{P}_{\boldsymbol{\lambda}}) + \log \frac{1}{\delta}) + K(\boldsymbol{\lambda})$.*

***Proof:*** In this proof, we extend our PAC-Bayes bound with data-independent priors to data-dependent ones that accommodate the error when the prior distribution is parameterized and optimized over a finite set of parameters $\mathfrak{P} = \{P_{\boldsymbol{\lambda}}, \boldsymbol{\lambda} \in \Lambda \subseteq \mathbb{R}^k\}$ with a much smaller dimension than the model itself. Let $\mathbb{T}(\Lambda, \|\cdot\|, \varepsilon)$ be an $\varepsilon$-cover of the set $\Lambda$, which states that for any $\boldsymbol{\lambda} \in \Lambda$, there exists a $\tilde{\boldsymbol{\lambda}} \in \mathbb{T}(\Lambda, \|\cdot\|, \varepsilon)$, such that $\|\boldsymbol{\lambda} - \tilde{\boldsymbol{\lambda}}\| \leq \varepsilon$.

Now we select the posterior distribution as $\mathcal{Q}_{\hat{\mathbf{w}}}$, parameterized by $\hat{\mathbf{w}} \in \mathbb{R}^d$. Assuming the prior $\mathcal{P}$ is parameterized by $\boldsymbol{\lambda} \in \mathbb{R}^k$ ($k \ll d$).

Then the PAC-Bayes bound 9 holds already for any $(\hat{\mathbf{w}}, \gamma, \boldsymbol{\lambda})$, with fixed $\boldsymbol{\lambda} \in \Lambda$ and $\gamma \in [\gamma_1, \gamma_2]$, i.e.,

$$\mathbb{E}_{\mathbf{h} \sim \mathcal{Q}_{\hat{\mathbf{w}}}} \ell(\mathbf{h}; \mathcal{D}) \leq \mathbb{E}_{\mathbf{h} \sim \mathcal{Q}_{\hat{\mathbf{w}}}} \ell(\mathbf{h}; \mathcal{S}) + \frac{1}{\gamma m} (\log \frac{1}{\delta} + \mathrm{KL}(\mathcal{Q}_{\hat{\mathbf{w}}} || \mathcal{P}_{\boldsymbol{\lambda}})) + \gamma K(\boldsymbol{\lambda}) \tag{18}$$

with probability over $1 - \delta$.

Now, for the collection of $\boldsymbol{\lambda}$s in the $\varepsilon$-net $\mathbb{T}(\Lambda, \|\cdot\|, \varepsilon)$, by the union bound, the PAC-Bayes bound uniformly holds on the $\varepsilon$-net with probability at least $1 - |\mathbb{T}|\delta = 1 - n(\varepsilon)\delta$. For an arbitrary $\boldsymbol{\lambda} \in \Lambda$, its distance to the $\varepsilon$-net is at most $\varepsilon$. Then under Assumption 4.13 and Assumption 4.14, we have:

$$\min_{\tilde{\boldsymbol{\lambda}} \in \mathbb{T}} |\mathrm{KL}(\mathcal{Q} || \mathcal{P}_{\boldsymbol{\lambda}}) - \mathrm{KL}(\mathcal{Q} || \mathcal{P}_{\tilde{\boldsymbol{\lambda}}})| \leq \eta_1(\|\boldsymbol{\lambda} - \tilde{\boldsymbol{\lambda}}\|) \leq \eta_1(\varepsilon),$$

and

$$\min_{\tilde{\boldsymbol{\lambda}} \in \mathbb{T}} |K(\boldsymbol{\lambda}) - K(\tilde{\boldsymbol{\lambda}})| \leq \eta_2(\|\boldsymbol{\lambda} - \tilde{\boldsymbol{\lambda}}\|) \leq \eta_2(\varepsilon).$$

Similarly, for $\gamma$, a $\varepsilon$-net on its range $\gamma_1 \leq \gamma \leq \gamma_2$ is the uniform grid with a grid separation $\varepsilon$, so the net contains $\frac{\gamma_2 - \gamma_1}{\varepsilon}$ points. By the union bound, requiring the PAC-Bayes bound to uniformly hold for all the $\gamma$ within this $\varepsilon$-net induces an extra probability of failure of $\frac{\gamma_2 - \gamma_1}{\varepsilon}\delta$. So, the total probability of failure is $n(\varepsilon)\delta + \frac{\gamma_2 - \gamma_1}{\varepsilon}\delta$.

For an arbitrary $\gamma \in \Gamma$, and $\Gamma := \{\gamma \in [\gamma_1, \gamma_2]\}$, its distance to the $\varepsilon$-net $\mathbb{T}'$ is at most $\varepsilon$, we have:

$$\min_{\tilde{\gamma} \in \mathbb{T}'} |L_{PAC}(\hat{\mathbf{w}}, \gamma, \boldsymbol{\lambda}, \delta) - L_{PAC}(\hat{\mathbf{w}}, \tilde{\gamma}, \boldsymbol{\lambda}, \delta)| = \frac{1}{m} \left( \mathrm{KL}(\mathcal{Q}_{\hat{\mathbf{w}}} || \mathcal{P}_{\boldsymbol{\lambda}}) + \log \frac{1}{\delta} \right) \left| \frac{1}{\gamma} - \frac{1}{\tilde{\gamma}} \right| + |\gamma - \tilde{\gamma}| K(\boldsymbol{\lambda})$$

$$= \left( \frac{1}{m\gamma\tilde{\gamma}} (\mathrm{KL}(\mathcal{Q}_{\hat{\mathbf{w}}} || \mathcal{P}_{\boldsymbol{\lambda}}) + \log \frac{1}{\delta}) + K(\boldsymbol{\lambda}) \right) |\gamma - \tilde{\gamma}|$$

$$\leq \left( \frac{1}{m\gamma_1^2} (\mathrm{KL}(\mathcal{Q}_{\hat{\mathbf{w}}} || \mathcal{P}_{\boldsymbol{\lambda}}) + \log \frac{1}{\delta}) + K(\boldsymbol{\lambda}) \right) \varepsilon$$

$$\leq B\varepsilon,$$

where $B := \sup_{\boldsymbol{\lambda} \in \Lambda} \frac{1}{m\gamma_1^2} (\mathrm{KL}(\mathcal{Q}_{\hat{\mathbf{w}}} || \mathcal{P}_{\boldsymbol{\lambda}}) + \log \frac{1}{\delta}) + K(\boldsymbol{\lambda})$, clearly, $B$ is a constant depending on the range of the parameters.

With the three inequalities above, we can control the PAC-Bayes loss at the given $\boldsymbol{\lambda}$ and $\gamma$ as follows:

$$
\min_{\tilde{\boldsymbol{\lambda}} \in \mathbb{T}, \tilde{\gamma} \in \mathbb{T}'} |L_{PAC}(\hat{\mathbf{w}}, \gamma, \boldsymbol{\lambda}, \delta) - L_{PAC}(\hat{\mathbf{w}}, \tilde{\gamma}, \tilde{\boldsymbol{\lambda}}, \delta)|
$$

$$
\leq \min_{\tilde{\gamma} \in \mathbb{T}'} |L_{PAC}(\hat{\mathbf{w}}, \gamma, \boldsymbol{\lambda}, \delta) - L_{PAC}(\hat{\mathbf{w}}, \tilde{\gamma}, \boldsymbol{\lambda}, \delta)| + \min_{\tilde{\boldsymbol{\lambda}} \in \mathbb{T}} |L_{PAC}(\hat{\mathbf{w}}, \tilde{\gamma}, \boldsymbol{\lambda}, \delta) - L_{PAC}(\hat{\mathbf{w}}, \tilde{\gamma}, \tilde{\boldsymbol{\lambda}}, \delta)|
$$

$$
\leq B\varepsilon + \frac{1}{\tilde{\gamma}m}\eta_1(\varepsilon) + \tilde{\gamma}\eta_2(\varepsilon)
$$

$$
\leq B\varepsilon + \frac{1}{\gamma_1 m}\eta_1(\varepsilon) + \gamma_2\eta_2(\varepsilon)
$$

$$
\leq B\varepsilon + C(\eta_1(\varepsilon) + \eta_2(\varepsilon))
$$

where $C = \frac{1}{\gamma_1} + \gamma_2$ and $\gamma_1 \leq \gamma \leq \gamma_2$. Since this inequality holds for any $\boldsymbol{\lambda} \in \Lambda$ and $\gamma \in \Gamma$, it certainly holds for the optima $\hat{\boldsymbol{\lambda}}$ and $\hat{\gamma}$. Combining this with Equation (18), we have

$$
\mathbb{E}_{\mathbf{h} \sim \mathcal{Q}_{\hat{\mathbf{w}}}} \ell(\mathbf{h}; \mathcal{D}) \leq L_{PAC}(\hat{\mathbf{w}}, \hat{\gamma}, \hat{\boldsymbol{\lambda}}, \delta) + B\varepsilon + C(\eta_1(\varepsilon) + \eta_2(\varepsilon)),
$$

where $B := \sup_{\boldsymbol{\lambda} \in \Lambda} \frac{1}{m\gamma_1^2}(\text{KL}(\mathcal{Q}_{\hat{\mathbf{w}}}||\mathcal{P}_{\boldsymbol{\lambda}}) + \log\frac{1}{\delta}) + K(\boldsymbol{\lambda})$.

Now taking $\epsilon := (n(\varepsilon) + \frac{\gamma_2 - \gamma_1}{\varepsilon})\delta$ to be the previously calculated probability of failure, we get, with probability $1 - \epsilon$, it holds that

$$
\mathbb{E}_{\mathbf{h} \sim \mathcal{Q}_{\hat{\mathbf{w}}}} \ell(\mathbf{h}; \mathcal{D}) \leq \mathbb{E}_{\mathbf{h} \sim \mathcal{Q}_{\hat{\mathbf{w}}}} \ell(\mathbf{h}; \mathcal{S}) + \frac{1}{\hat{\gamma}m}\left[\log\frac{n(\varepsilon) + \frac{\gamma_2 - \gamma_1}{\varepsilon}}{\epsilon} + \text{KL}(\mathcal{Q}_{\hat{\mathbf{w}}}||\mathcal{P}_{\hat{\boldsymbol{\lambda}}})\right] + \hat{\gamma}K(\hat{\boldsymbol{\lambda}}) + B\varepsilon + C(\eta_1(\varepsilon) + \eta_2(\varepsilon))
$$

$$
\leq L_{PAC}(\hat{\mathbf{w}}, \hat{\gamma}, \hat{\boldsymbol{\lambda}}, \delta) + \eta \tag{19}
$$

and the proof is completed. $\qquad\square$

### A.3 KL divergence of the Gaussian prior and posterior

For a $k$-layer network, the prior is written as $\mathcal{P}_{\boldsymbol{\mu}_0, \boldsymbol{\lambda}}$, where $\boldsymbol{\mu}_0$ is the random initialized model parameter and $\boldsymbol{\lambda} \in \mathbb{R}_+^k$ is the vector containing the variance for each layer. The set of all such priors is denoted by $\mathfrak{P} := \{\mathcal{P}_{\boldsymbol{\mu}_0, \boldsymbol{\lambda}}, \boldsymbol{\lambda} \in \Lambda \subseteq \mathbb{R}^k, \boldsymbol{\mu}_0 \in \Theta\}$. In the PAC-Bayes training, we select the posterior distribution to be centered around the trained model parameterized by $\boldsymbol{\mu}$, with independent anisotropic variance. Specifically, for a network with $d$ trainable parameters, the posterior is $\mathcal{Q}_{\boldsymbol{\mu}, \boldsymbol{\sigma}} := \mathcal{N}(\boldsymbol{\mu}, \text{diag}(\boldsymbol{\sigma}))$, where $\boldsymbol{\mu}$ (the current model) is the mean and $\boldsymbol{\sigma} \in \mathbb{R}_+^d$ is the vector containing the variance for each trainable parameter. The set of all posteriors is $\mathfrak{Q} := \{\mathcal{Q}_{\boldsymbol{\mu}, \boldsymbol{\sigma}}, \boldsymbol{\sigma} \in \Sigma, \boldsymbol{\mu} \in \Theta\}$, and the KL divergence between all such prior and posterior in $\mathfrak{P}$ and $\mathfrak{Q}$ is:

$$
\text{KL}(\mathcal{Q}_{\boldsymbol{\mu}, \boldsymbol{\sigma}}||\mathcal{P}_{\boldsymbol{\mu}_0, \boldsymbol{\lambda}}) = \frac{1}{2}\sum_{i=1}^{k}\left[-\mathbf{1}_{d_i}^\top \log(\boldsymbol{\sigma}_i) + d_i(\log(\boldsymbol{\lambda}_i) - 1) + \frac{\|\boldsymbol{\sigma}_i\|_1 + \|(\boldsymbol{\mu} - \boldsymbol{\mu}_0)_i\|_2^2}{\boldsymbol{\lambda}_i}\right], \tag{20}
$$

where $\boldsymbol{\sigma}_i, (\boldsymbol{\mu} - \boldsymbol{\mu}_0)_i$ are vectors denoting the variances and weights for the $i$-th layer, respectively, and $\lambda_i$ is the scalar variance for the $i$-th layer. $d_i = \dim(\boldsymbol{\sigma}_i)$, and $\mathbf{1}_{d_i}$ denotes an all-ones vector of length $d_i$[5].

Scalar prior is a special case of the layerwise prior by setting all entries of $\boldsymbol{\lambda}$ to be equal, for which the KL divergence reduces to

$$
\text{KL}(\mathcal{Q}_{\boldsymbol{\mu}, \boldsymbol{\sigma}}||\mathcal{P}_{\boldsymbol{\mu}_0, \lambda}) = \frac{1}{2}\left[-\mathbf{1}_d^\top \log(\boldsymbol{\sigma}) + d(\log(\lambda) - 1) + \frac{1}{\lambda}(\|\boldsymbol{\sigma}\|_1 + \|\boldsymbol{\mu} - \boldsymbol{\mu}_0\|_2^2)\right]. \tag{21}
$$

---

[5]Note that with a little ambiguity, the $\boldsymbol{\lambda}_i$ here has a different meaning from that in Equation (26) and Algorithm 2, here $\boldsymbol{\lambda}_i$ means the $i$th element in $\boldsymbol{\lambda}$, whereas in Equation (26) and Algorithm 2, $\boldsymbol{\lambda}_i$ means the $i$th element in the discrete set.

## A.4 Proof of Corollary 4.16

Recall for the training, we proposed to optimize over all four variables: $\boldsymbol{\mu}$, $\gamma$, $\boldsymbol{\sigma}$, and $\boldsymbol{\lambda}$.

$$(\hat{\boldsymbol{\mu}}, \hat{\boldsymbol{\sigma}}, \hat{\gamma}, \hat{\boldsymbol{\lambda}}) = \arg \min_{\substack{\boldsymbol{\mu}, \boldsymbol{\lambda}, \boldsymbol{\sigma}, \\ \gamma \in [\gamma_1, \gamma_2]}} \underbrace{\mathbb{E}_{\boldsymbol{\theta} \sim \mathcal{Q}_{\boldsymbol{\mu}, \boldsymbol{\sigma}}} \ell(f_{\boldsymbol{\theta}}; \mathcal{S}) + \frac{1}{\gamma m}(\log \frac{1}{\delta} + \mathrm{KL}(\mathcal{Q}_{\boldsymbol{\mu}, \boldsymbol{\sigma}} || \mathcal{P}_{\boldsymbol{\mu}_0, \boldsymbol{\lambda}})) + \gamma K(\boldsymbol{\lambda})}_{\equiv L_{PAC}([\boldsymbol{\mu}, \boldsymbol{\sigma}], \gamma, \boldsymbol{\lambda}, \delta)}. \tag{22}$$

**Corollary A.3.** *Assume all parameters for the prior and posterior are bounded, i.e., we restrict the model parameter $\boldsymbol{\mu}$, the posterior variance $\boldsymbol{\sigma}$ and the prior variance $\boldsymbol{\lambda}$, and the exponential moment $K(\boldsymbol{\lambda})$ all to be searched over bounded sets, $\Theta := \{\boldsymbol{\mu} \in \mathbb{R}^d : \|\boldsymbol{\mu}\|_2 \leq \sqrt{d}M\}$, $\Sigma := \{\boldsymbol{\sigma} \in \mathbb{R}_+^d : \|\boldsymbol{\sigma}\|_1 \leq dT\}$, $\Lambda =: \{\boldsymbol{\lambda} \in [e^{-a}, e^b]^k\}$, $\Gamma := \{\gamma \in [\gamma_1, \gamma_2]\}$, respectively, with fixed $M, T, a, b > 0$. Then,*

- *Assumption 4.13 holds with $\eta_1(x) = L_1 x$, where $L_1 = \frac{1}{2} \max\{d, e^a(2\sqrt{d}M + dT)\}$*

- *Assumption 4.14 holds with $\eta_2(x) = L_2 x$, where $L_2 = \frac{1}{\gamma_1^2}\left(2dM^2 e^{2a} + \frac{d(a+b)}{2}\right)$*

- *With high probability, the PAC-Bayes bound for the minimizer of Equation (P) has the form*

$$\mathbb{E}_{\boldsymbol{\theta} \sim \mathcal{Q}_{\hat{\boldsymbol{\mu}}, \hat{\boldsymbol{\sigma}}}} \ell(f_{\boldsymbol{\theta}}; \mathcal{D}) \leq L_{PAC}([\hat{\boldsymbol{\mu}}, \hat{\boldsymbol{\sigma}}], \hat{\gamma}, \hat{\boldsymbol{\lambda}}, \delta) + \eta,$$

*where $\eta = \frac{k}{\gamma_1 m}\left(1 + \log \frac{2(CL+B)\Delta\gamma_1 m}{k}\right)$, $L = L_1 + L_2$, $\Delta := \max\{b + a, 2(\gamma_2 - \gamma_1)\}$, $B = \sup_{\boldsymbol{\lambda} \in \Lambda} \frac{1}{m\gamma_1^2}(\mathrm{KL}(\mathcal{Q}_{\hat{\boldsymbol{\mu}}, \hat{\boldsymbol{\sigma}}} || \mathcal{P}_{\boldsymbol{\mu}_0, \boldsymbol{\lambda}}) + \log \frac{1}{\delta}) + K(\boldsymbol{\lambda})$, and $C = \frac{1}{\gamma_1 m} + \gamma_2$.*

***Proof:*** We first prove the two assumptions are satisfied by the Gaussian family with bounded parameter spaces. To prove Assumption 4.13 is satisfied, let $v_i = \log 1/\lambda_i$, $i = 1, ..., k$ and perform a change of variable from $\lambda_i$ to $v_i$. The weight of prior for the $i$th layer now becomes $\mathcal{N}(\boldsymbol{\mu}_0, e^{-v_i}\mathbf{I}_{d_i}))$, where $d_i$ is the number of trainable parameters in the $i$th layer. It is straightforward to compute

$$\frac{\partial \mathrm{KL}(\mathcal{Q}_{\boldsymbol{\mu}, \boldsymbol{\sigma}} || \tilde{\mathcal{P}}_{\boldsymbol{\mu}_0, \mathbf{v}})}{\partial v_i} = \frac{1}{2}[-d_i + e^{v_i}(\|\boldsymbol{\sigma}_i\|_1 + \|\boldsymbol{\mu}_i - \boldsymbol{\mu}_{0,i}\|_2^2)],$$

where $\boldsymbol{\sigma}_i$, $\boldsymbol{\mu}_i$, $\boldsymbol{\mu}_{0,i}$ are the blocks of $\boldsymbol{\sigma}$, $\boldsymbol{\mu}$, $\boldsymbol{\mu}_0$, containing the parameters associated with the $i$th layer, respectively. Now, given the assumptions on the boundedness of the parameters, we have:

$$\|\nabla_{\mathbf{v}} \mathrm{KL}(\mathcal{Q}_{\boldsymbol{\mu}, \boldsymbol{\sigma}} || \tilde{\mathcal{P}}_{\boldsymbol{\mu}_0, \mathbf{v}})\|_2 \leq \|\nabla_{\mathbf{v}} \mathrm{KL}(\mathcal{Q}_{\boldsymbol{\mu}, \boldsymbol{\sigma}} || \tilde{\mathcal{P}}_{\boldsymbol{\mu}_0, \mathbf{v}})\|_1 \leq \frac{1}{2} \max\{d, e^a(2\sqrt{d}M + dT)\} \equiv L_1(d, M, T, a), \tag{23}$$

where we used the assumption $\|\boldsymbol{\sigma}\|_1 \leq dT$ and $\|\boldsymbol{\mu}_0\|_2, \|\boldsymbol{\mu}\|_2 \leq \sqrt{d}M$.

Equation 23 says $L_1(d, M, T, a)$ is a valid Lipschitz bound on the KL divergence and therefore Assumption 4.13 is satisfied by setting $\eta_1(x) = L_1(d, M, T, a)x$.

Next, we prove Assumption 4.14 is satisfied. We use $K_{\min}(\boldsymbol{\lambda})$ defined in Definition 4.7 as the $K(\boldsymbol{\lambda})$ in the PAC-Bayes training, and verify that it makes Assumption 4.14 hold.

$$|K_{\min}(\boldsymbol{\lambda}_1) - K_{\min}(\boldsymbol{\lambda}_2)|$$

$$= \left| \sup_{\gamma \in [\gamma_1, \gamma_2]} \frac{1}{\gamma^2} \log(\mathbb{E}_{\boldsymbol{\theta} \sim \mathcal{P}_{\boldsymbol{\mu}_0, \boldsymbol{\lambda}_1}} \mathbb{E}_{z \sim \mathcal{D}}[\exp(\gamma \ell(f_{\boldsymbol{\theta}}; z))]) - \sup_{\gamma \in [\gamma_1, \gamma_2]} \frac{1}{\gamma^2} \log(\mathbb{E}_{\boldsymbol{\theta} \sim \mathcal{P}_{\boldsymbol{\mu}_0, \boldsymbol{\lambda}_2}} \mathbb{E}_{z \sim \mathcal{D}}[\exp(\gamma \ell(f_{\boldsymbol{\theta}}; z))]) \right|$$

$$\leq \sup_{\gamma \in [\gamma_1, \gamma_2]} \frac{1}{\gamma^2} \left| \log(\mathbb{E}_{\boldsymbol{\theta} \sim \mathcal{P}_{\boldsymbol{\mu}_0, \boldsymbol{\lambda}_1}} \mathbb{E}_{z \sim \mathcal{D}}[\exp(\gamma \ell(f_{\boldsymbol{\theta}}; z))]) - \log(\mathbb{E}_{\boldsymbol{\theta} \sim \mathcal{P}_{\boldsymbol{\mu}_0, \boldsymbol{\lambda}_2}} \mathbb{E}_{z \sim \mathcal{D}}[\exp(\gamma \ell(f_{\boldsymbol{\theta}}; z))]) \right|$$

$$= \sup_{\gamma \in [\gamma_1, \gamma_2]} \frac{1}{\gamma^2} \left| \log(\mathbb{E}_{\boldsymbol{\theta} \sim \mathcal{P}_{\boldsymbol{\mu}_0, \boldsymbol{\lambda}_2}} \mathbb{E}_{z \sim \mathcal{D}}[\exp(\gamma \ell(f_{\boldsymbol{\theta}}; z))] \frac{p_{\boldsymbol{\mu}_0, \boldsymbol{\lambda}_1}(\boldsymbol{\theta})}{p_{\boldsymbol{\mu}_0, \boldsymbol{\lambda}_2}(\boldsymbol{\theta})}) - \log(\mathbb{E}_{\boldsymbol{\theta} \sim \mathcal{P}_{\boldsymbol{\mu}_0, \boldsymbol{\lambda}_2}} \mathbb{E}_{z \sim \mathcal{D}}[\exp(\gamma \ell(f_{\boldsymbol{\theta}}; z))]) \right|$$

$$\leq \sup_{\gamma \in [\gamma_1, \gamma_2]} \frac{1}{\gamma^2} \sup_{\boldsymbol{\theta} \in \Theta} \left| \log \frac{p_{\boldsymbol{\mu}_0, \boldsymbol{\lambda}_1}(\boldsymbol{\theta})}{p_{\boldsymbol{\mu}_0, \boldsymbol{\lambda}_2}(\boldsymbol{\theta})} \right|$$

$$\leq \frac{1}{\gamma_1^2} \sup_{\mathbf{h} \in \mathcal{H}} \left| \log \frac{p_{\boldsymbol{\mu}_0, \boldsymbol{\lambda}_1}(\boldsymbol{\theta})}{p_{\boldsymbol{\mu}_0, \boldsymbol{\lambda}_2}(\boldsymbol{\theta})} \right|$$

$$\leq \frac{1}{\gamma_1^2} \left( 2dM^2 e^{2a} + \frac{d(a+b)}{2} \right) \|\boldsymbol{\lambda}_1 - \boldsymbol{\lambda}_2\|_2,$$

where the first inequality used the property of the supremum, the $p_{\boldsymbol{\mu}_0, \boldsymbol{\lambda}_1}(\boldsymbol{\theta}), p_{\boldsymbol{\mu}_0, \boldsymbol{\lambda}_2}(\boldsymbol{\theta})$ in the fourth line denote the probability density function of Gaussian with mean $\boldsymbol{\mu}_0$ and variance parametrized by $\boldsymbol{\lambda}_1, \boldsymbol{\lambda}_2$ (i.e., $\boldsymbol{\lambda}_{1,i}, \boldsymbol{\lambda}_{2,i}$ are the variances for the $i$th layer), the second inequality use the fact that if $X(\mathbf{h})$ is a non-negative function of $\mathbf{h}$ and $Y(\mathbf{h})$ is a bounded function of $\mathbf{h}$, then

$$|\mathbb{E}_{\mathbf{h}}(X(\mathbf{h})Y(\mathbf{h}))| \leq (\sup_{\mathbf{h} \in \mathcal{H}} |Y(\mathbf{h})|) \cdot \mathbb{E}_{\mathbf{h}} X(\mathbf{h}).$$

The last inequality used the formula of the Gaussian density

$$p(x; \mu, \Sigma) = \frac{1}{(2\pi)^{d/2} |\Sigma|^{1/2}} \exp\left( -\frac{1}{2}(x - \mu)^T \Sigma^{-1}(x - \mu) \right)$$

and the boundedness of the parameters. Therefore, Assumption 4.14 is satisfied by setting $\eta_2(x) = L_2(d, M, \gamma_1, a)x$, where $L_2(d, M, \gamma_1, a) = \frac{1}{\gamma_1^2} \left( 2dM^2 e^{2a} + \frac{d(a+b)}{2} \right)$.

Let $L(d, M, T, \gamma_1, a) = L_1(d, M, T, a) + L_2(d, M, \gamma_1, a)$. Then we can apply Theorem 4.15, to get with probability $1 - \epsilon$,

$$\mathbb{E}_{\boldsymbol{\theta} \sim \mathcal{Q}_{\hat{\boldsymbol{\mu}}, \hat{\boldsymbol{\sigma}}}} \ell(f_{\boldsymbol{\theta}}; \mathcal{D})$$

$$\leq \mathbb{E}_{\boldsymbol{\theta} \sim \mathcal{Q}_{\hat{\boldsymbol{\mu}}, \hat{\boldsymbol{\sigma}}}} \ell(f_{\boldsymbol{\theta}}; \mathcal{S}) + \frac{1}{\hat{\gamma}m} \left[ \log \frac{n(\varepsilon) + \frac{\gamma_2 - \gamma_1}{2\varepsilon}}{\epsilon} + \mathrm{KL}(\mathcal{Q}_{\hat{\boldsymbol{\mu}}, \hat{\boldsymbol{\sigma}}} || \mathcal{P}_{\boldsymbol{\mu}_0, \hat{\boldsymbol{\lambda}}}) \right] + \hat{\gamma} K_{\min}(\hat{\boldsymbol{\lambda}}) + \tag{24}$$

$$(CL(d, M, T, \gamma_1, a)) + B)\varepsilon.$$

Here, we used $\eta_1(x) = L_1 x$ and $\eta_2(x) = L_2 x$. Note that for the set $[-b, a]^k$, the covering number $n(\varepsilon) = \mathcal{N}([-b, a]^k, |\cdot|, \varepsilon)$ is $\left( \frac{b+a}{2\varepsilon} \right)^k$, and the covering number $\frac{\gamma_2 - \gamma_1}{2\varepsilon}$ for $\gamma \in [\gamma_1, \gamma_2]$.

We introduce a new variable $\rho > 0$, letting $\varepsilon = \frac{\rho}{2(CL(d, M, T, \gamma_1, a) + B)}$ and inserting it into Equation (24), we obtain with probability $1 - \epsilon$:

$$\mathbb{E}_{\boldsymbol{\theta} \sim \mathcal{Q}_{\hat{\boldsymbol{\mu}}, \hat{\boldsymbol{\sigma}}}} \ell(f_{\boldsymbol{\theta}}; \mathcal{D})$$

$$\leq \mathbb{E}_{\boldsymbol{\theta} \sim \mathcal{Q}_{\hat{\boldsymbol{\mu}}, \hat{\boldsymbol{\sigma}}}} \ell(f_{\boldsymbol{\theta}}; \mathcal{S}) + \frac{1}{\hat{\gamma}m} \left[ \log \frac{1}{\epsilon} + \mathrm{KL}(\mathcal{Q}_{\hat{\boldsymbol{\mu}}, \hat{\boldsymbol{\sigma}}} || \mathcal{P}_{\boldsymbol{\mu}_0, \hat{\boldsymbol{\lambda}}}) \right]$$

$$+ \hat{\gamma} K_{\min}(\hat{\boldsymbol{\lambda}}) + \rho + \frac{k}{\gamma_1 m} \log \frac{2(CL(d, M, T, \gamma_1, a) + B)\Delta}{\rho}.$$

where $\Delta := \max\{b + a, 2(\gamma_2 - \gamma_1)\}$.

Optimizing over $\rho$, we obtain:

$$\mathbb{E}_{\boldsymbol{\theta} \sim \mathcal{Q}_{\hat{\boldsymbol{\mu}}, \hat{\boldsymbol{\sigma}}}} \ell(f_{\boldsymbol{\theta}}; \mathcal{D})$$

$$\leq \mathbb{E}_{\boldsymbol{\theta} \sim \mathcal{Q}_{\hat{\boldsymbol{\mu}}, \hat{\boldsymbol{\sigma}}}} \ell(f_{\boldsymbol{\theta}}; \mathcal{S}) + \frac{1}{\hat{\gamma} m} \left[ \log \frac{1}{\epsilon} + \mathrm{KL}(\mathcal{Q}_{\hat{\boldsymbol{\mu}}, \hat{\boldsymbol{\sigma}}} || \mathcal{P}_{\boldsymbol{\mu}_0, \hat{\boldsymbol{\lambda}}}) \right]$$

$$+ \hat{\gamma} K_{\min}(\hat{\boldsymbol{\lambda}}) + \frac{k}{\gamma_1 m} \left( 1 + \log \frac{2(CL(d, M, T, \gamma_1, a) + B) \Delta \gamma_1 m}{k} \right)$$

$$= L_{PAC}([\hat{\boldsymbol{\mu}}, \hat{\boldsymbol{\sigma}}], \hat{\gamma}, \hat{\boldsymbol{\lambda}}, \delta) + \frac{k}{\gamma_1 m} \left( 1 + \log \frac{2(CL(d, M, T, \gamma_1, a) + B) \Delta \gamma_1 m}{k} \right).$$

Hence we have

$$\mathbb{E}_{\boldsymbol{\theta} \sim \mathcal{Q}_{\hat{\boldsymbol{\mu}}, \hat{\boldsymbol{\sigma}}}} \ell(f_{\boldsymbol{\theta}}; \mathcal{D}) \leq L_{PAC}([\hat{\boldsymbol{\mu}}, \hat{\boldsymbol{\sigma}}], \hat{\gamma}, \hat{\boldsymbol{\lambda}}, \delta) + \eta,$$

where $\eta = \max \left( \frac{1}{\gamma_1 m}(1 + \log(2(CL(d, M, T, \gamma_1, a) + B)(\gamma_2 - \gamma_1)\gamma_1 m)), \frac{k}{\gamma_1 m} \left( 1 + \log \frac{2(CL(d, M, T, \gamma_1, a) + B)\Delta \gamma_1 m}{k} \right) \right)$.

$\square$

*Remark* A.4. In defining the boundedness of the domain $\Theta$ of $\boldsymbol{\mu}$ in Corollary 4.16, we used $\sqrt{d}M$ as the bound. Here, the factor $\sqrt{d}$ (where $d$ denotes the dimension of $\mathbf{h}$) is used to encapsulate the idea that if on average, the components of the weight are bounded by $M$, then the $\ell_2$ norm would naturally be bounded by $\sqrt{d}M$. The same idea applies to the definition of $\Sigma$.

*Remark* A.5. Due to the above remark, $M$, $T$, $a$, $b$ can be treated as dimension-independent constants that do not grow with the network size $d$. As a result, the constants $L_1, L_2, L$ in Corollary 4.16, are dominated by $d$, and $L_1, L_2, L = O(d)$. This then implies the logarithm term in $\eta$ scales as $O(\log d)$, which grows very mildly with the size. Therefore, Corollary 4.16 can be used as the generalization guarantee for large neural networks.

# B  Algorithm Details

## B.1  Algorithms to estimate $K(\lambda)$

In this section, we explain the algorithm to compute $K(\lambda)$. In previous literature, the moment bound $K$ or its analog term in the PAC-Bayes bounds was often assumed to be a constant. One of our contributions is to allow $K$ to vary with the variance $\lambda$ of the prior, so if a small prior variance is found by PAC-Bayes training, then the corresponding $K$ would also be small. We perform linear interpolation to approximate the function $K_{\min}(\lambda)$ defined in (2) of the main text. When $\lambda$ is 1D, We first compute $K_{\min}(\lambda)$ on a finite grid of the domain of $\lambda$, by solving Equation (25) below. With the computed function values on the grid $\{K_{\min}(\lambda_i)\}_i$, we can construct a piecewise linear function as the approximation of $K_{\min}(\lambda)$.

$$K_{\min}(\lambda_i) = \arg \min_{K > 0} K$$

$$\text{s.t.} \exp\left(\gamma^2 K\right) \geq \frac{1}{nm} \sum_{l=1}^{n} \sum_{j=1}^{m} \exp(\gamma(\ell(f_{\boldsymbol{\theta}_l}; \mathcal{S}) - \ell(f_{\boldsymbol{\theta}_l}; z_j))), \tag{25}$$

$$\forall \, \gamma \in [\gamma_1, \gamma_2], \quad \boldsymbol{\theta}_l \sim \mathcal{N}(\boldsymbol{\mu}_0, \lambda_i), \quad \lambda_{\min} \leq \lambda_i \leq \lambda_{\max}$$

where $\boldsymbol{\theta}_l \sim \mathcal{P}_{\boldsymbol{\mu}_0, \lambda_i}, l = 1, ..., n$, are samples from the prior distribution and are fixed when solving Equation (25) for $K_{\min}(\boldsymbol{\lambda}_i)$. Equation (25) is the discrete version of the formula (2) in the main text. This optimization problem is 1-dimensional, and the function in the constraint is monotonic in $K$, so it can be solved efficiently by the bisection method.

When extending this procedure to high dimension, where $\boldsymbol{\lambda}$ is a $k$-dimension vector, we need to set up a grid for the domain of $\boldsymbol{\lambda}$ in $k$-dimensional space and estimate $K_{\min}$ on each grid point, which is time-consuming when $k$ is large. To address this issue, we propose to use the following approximation:

$$\hat{K}(\max(\boldsymbol{\lambda}_i)) = \arg\min_{K>0} K$$

$$\text{s.t. } \exp\left(\gamma^2 K\right) \geq \frac{1}{nm} \sum_{l=1}^{n} \sum_{j=1}^{m} \exp(\gamma(\ell(f_{\boldsymbol{\theta}_i}; \mathcal{S}) - \ell(f_{\boldsymbol{\theta}_i}; z_j))), \tag{26}$$

$$\forall\, \gamma \in [\gamma_1, \gamma_2], \quad \boldsymbol{\theta}_l \sim \mathcal{N}(\boldsymbol{\mu}_0, \max(\boldsymbol{\lambda}_i)), \quad \lambda_{\min} \leq \max(\boldsymbol{\lambda}_i) \leq \lambda_{\max}, i = 1, ..., s$$

where $\boldsymbol{\lambda}_i$ is a random sample from the domain $\Lambda$ of $\boldsymbol{\lambda}$. Since each $\boldsymbol{\lambda_i}$ is k-dimensional, $\max(\boldsymbol{\lambda_i})$ represents the maximum of the $k$ coordinates.

The idea of this formulation 26 is as follows, we use the 1D function $\hat{K}(\max(\boldsymbol{\lambda}_i))$ as a surrogate function of the original $k$-dimension function $K_{\min}(\boldsymbol{\lambda})$ (i.e. $K_{\min}(\boldsymbol{\lambda}) \leq \hat{K}(\max(\boldsymbol{\lambda}_i))$). Then estimating this 1D surrogate function is easy by using the bisection method. This procedure will certainly overestimate the true $K_{\min}(\boldsymbol{\lambda})$ but since the surrogate function is also a valid exponential moment bound, it is safe to be used as a replacement for the $K(\boldsymbol{\lambda})$ in our PAC-Bayes bound for training. In practice, we tried to use $\text{mean}(\boldsymbol{\lambda}_i)$ to replace $\max(\boldsymbol{\lambda}_i)$ to mitigate the over-estimation, but the final performance stays the same. The details of the whole procedure are presented in Algorithm 2.

---

**Algorithm 2** Compute $K(\boldsymbol{\lambda})$ given a set of query priors

**Input:** $\gamma_1$ and $\gamma_2$, sampling time $s$ of prior variances, the initial neural network weight $\boldsymbol{\theta}_0$, the training dataset $\mathcal{S} = \{z_i\}_{i=1}^{m}$, model sampling time $n = 10$
**Output:** the piece-wise linear interpolation $\tilde{K}(\boldsymbol{\lambda})$ for $K_{\min}(\boldsymbol{\lambda})$
Draw $s$ random samples for the prior variances $\mathcal{V} = \{\boldsymbol{\lambda}_i \in \Lambda \subseteq \mathbb{R}^k, i = 1, ..., s\}$
Set up a discrete grid $\Gamma$ for the interval $[\gamma_1, \gamma_2]$ of $\gamma$.
**for** $\boldsymbol{\lambda}_i \in \mathcal{V}$ **do**
  **for** $l = 1 : n$ **do**
    Sampling weights from the Gaussian distribution $\boldsymbol{\theta}_l \sim \mathcal{N}(\boldsymbol{\mu}_0, \boldsymbol{\lambda}_i)$
    Use $\boldsymbol{\theta}_l$, $\Gamma$ and $\mathcal{S}$ to compute one term in the sum in Equation (26)
  **end for**
  Solve $\hat{K}(\max(\boldsymbol{\lambda}_i))$ using Equation (26)
**end for**
Fit a piece-wise linear function $\tilde{K}(\boldsymbol{\lambda})$ to the data $\{(\boldsymbol{\lambda}_i, \hat{K}(\max(\boldsymbol{\lambda}_i)))\}_{i=1}^{s}$

---

## B.2 PAC-Bayes Training with layerwise prior

Similar to Algorithm 1, our PAC-Bayes training with a layerwise prior is stated here in Algorithm 3.

## B.3 Regularizations in PAC-Bayes bound

Only noise injection and weight decay are essential from our derived PAC-Bayes bound. Since many factors in normal training, such as mini-batch and dropout, enhance generalization by some sort of noise injection, it is unsurprising that they can be substituted by the well-calibrated noise injection in PAC-Bayes training. Like most commonly used implicit regularizations (large lr, momentum, small batch size), dropout and batch-norm are also known to penalize the loss function's sharpness indirectly. Wei et al. (2020) studies that dropout introduces an explicit regularization that penalizes *sharpness* and an implicit regularization that is analogous to the effect of stochasticity in small mini-batch stochastic gradient descent. Similarly, it is well-studied that batch-norm Luo et al. (2018) allows the use of a large learning rate by reducing the variance in the layer batches, and large allowable learning rates regularize *sharpness* through the edge of stability Cohen et al. (2020). As shown in the equation below, the first term (noise-injection) in our PAC-Bayes bound explicitly penalizes the Trace of the Hessian of the loss, which directly relates to sharpness and is quite similar

---

**Algorithm 3** PAC-Bayes training (layerwise prior)

---

**Input:** initial weight $\boldsymbol{\mu}_0 \in \mathbb{R}^d$, the number of layers $k$, $T_1$, $\lambda_1 = e^{-12}, \lambda_2 = e^2$, $\gamma_1 = 0.5, \gamma_2 = 10$, //
$T_1, \lambda_1, \lambda_2, \gamma_1, \gamma_2$ *can be fixed in all experiments of Sec5.*
**Output:** trained model $\hat{\boldsymbol{\mu}}$, posterior noise level $\hat{\boldsymbol{\sigma}}$
$\boldsymbol{\theta} \leftarrow \boldsymbol{\mu}_0$, $\mathbf{v} \leftarrow \mathbf{1}_d \cdot \log(\frac{1}{d}\sum_{i=1}^d |\boldsymbol{\mu}_{0,i}|)$, $\mathbf{b} \leftarrow \mathbf{1}_k \cdot \log(\frac{1}{d}\sum_{i=1}^d |\boldsymbol{\mu}_{0,i}|)$              *// Initialization*
Obtain the estimated $\tilde{K}(\bar{\boldsymbol{\lambda}})$ with $\Lambda = [\lambda_1, \lambda_2]^k$ using Equation (26) and Appendix B.1
*// Stage 1*
**for** epoch $= 1 : T_1$ **do**
   **for** sampling one batch $s$ from $\mathcal{S}$ **do**
      $\boldsymbol{\lambda} \leftarrow \exp(\mathbf{b})$, $\boldsymbol{\sigma} \leftarrow \exp(\mathbf{v})$                 *// Ensure non-negative variances*
      Construct the covariance of $\mathcal{P}_{\hat{\boldsymbol{\mu}}_0, \boldsymbol{\lambda}}$ from $\boldsymbol{\lambda}$   *// Setting the variance of the weights in layer-i all to the scalar $\boldsymbol{\lambda}(i)$*
      Draw one $\tilde{\boldsymbol{\theta}} \sim \mathcal{Q}_{\boldsymbol{\mu},\boldsymbol{\sigma}}$ and evaluate $\ell(f_{\tilde{\boldsymbol{\theta}}}; \mathcal{S})$,        *// Stochastic version of $\mathbb{E}_{\tilde{\boldsymbol{\theta}} \sim \mathcal{Q}_{\boldsymbol{\mu},\boldsymbol{\sigma}}} \ell(f_{\tilde{\boldsymbol{\theta}}}; \mathcal{S})$*
      Compute the KL-divergence as Equation (20)
      Compute $\gamma$ as Equation (8)
      Compute the loss function $\mathcal{L}$ as $L_{PAC}$ in Equation (P)
      $\mathbf{b} \leftarrow \mathbf{b} + \eta \frac{\partial \mathcal{L}}{\partial \mathbf{b}}$, $\mathbf{v} \leftarrow \mathbf{v} + \eta \frac{\partial \mathcal{L}}{\partial \mathbf{v}}$, $\boldsymbol{\mu} \leftarrow \boldsymbol{\mu} + \eta \frac{\partial \mathcal{L}}{\partial \boldsymbol{\mu}}$         *// Update all parameters*
   **end for**
**end for**
$\hat{\boldsymbol{\sigma}} \leftarrow \exp(\mathbf{v})$                        *// Fix the noise level from now on*
*// Stage 2*
**while** not converge **do**
   **for** sampling one batch $s$ from $\mathcal{S}$ **do**
      Draw one sample $\tilde{\boldsymbol{\theta}} \sim \mathcal{Q}_{\hat{\boldsymbol{\mu}},\hat{\boldsymbol{\sigma}}}$ and evaluate $\ell(f_{\tilde{\boldsymbol{\theta}}}; \mathcal{S})$ as $\tilde{\mathcal{L}}$,        *// Noise injection*
      $\boldsymbol{\mu} \leftarrow \boldsymbol{\mu} + \eta \frac{\partial \tilde{\mathcal{L}}}{\partial \boldsymbol{\mu}}$               *// Update model parameters*
   **end for**
**end while**
$\hat{\boldsymbol{\mu}} \leftarrow \boldsymbol{\mu}$

---

to the regularization effect of batch-norm and dropout. During training, suppose the current posterior is $\mathcal{Q}_{\hat{\boldsymbol{\mu}},\hat{\boldsymbol{\sigma}}} = \mathcal{N}(\hat{\boldsymbol{\mu}}, \text{diag}(\hat{\boldsymbol{\sigma}}))$, then the training loss expectation over the posterior is:

$$\mathbb{E}_{\boldsymbol{\theta} \sim \mathcal{Q}_{\hat{\boldsymbol{\mu}},\hat{\boldsymbol{\sigma}}}} \ell(f_{\boldsymbol{\theta}}; \mathcal{D}) = \mathbb{E}_{\Delta\boldsymbol{\theta} \sim \mathcal{Q}_{\mathbf{0},\hat{\boldsymbol{\sigma}}}} \ell(f_{\hat{\boldsymbol{\theta}}+\Delta\boldsymbol{\theta}}; \mathcal{D})$$

$$\approx \ell(f_{\hat{\boldsymbol{\theta}}}, \mathcal{D}) + \mathbb{E}_{\Delta\boldsymbol{\theta} \sim \mathcal{Q}_{\mathbf{0},\hat{\boldsymbol{\sigma}}}} (\ell(f_{\hat{\boldsymbol{\theta}}}; \mathcal{D})\Delta\boldsymbol{\theta} + \frac{1}{2}\Delta\boldsymbol{\theta}^\top \nabla^2 \ell(f_{\hat{\boldsymbol{\theta}}}; \mathcal{D})\Delta\boldsymbol{\theta})$$

$$= \ell(\hat{f}_{\boldsymbol{\theta}}; \mathcal{D}) + \frac{1}{2}\text{Tr}(\text{diag}(\hat{\boldsymbol{\sigma}})\nabla^2 \ell(f_{\hat{\boldsymbol{\theta}}}; \mathcal{D})).$$

The second regularization term (weight decay) in the bound additionally ensures that the minimizer found is close to initialization. Although the relation of this regularizer to sharpness is not very clear, empirical results suggest that weight decay may have a separate regularization effect from sharpness. In brief, we state that the effect of sharpness regularization from dropout and batch norm can also be well emulated by noise injection with the additional effect of weight decay.

### B.4 Deterministic Prediction

Recall that for any $\boldsymbol{\mu} \in \mathbb{R}^d$ and $\boldsymbol{\sigma} \in \mathbb{R}^d_+$, we used $\mathcal{Q}_{\boldsymbol{\mu},\boldsymbol{\sigma}}$ to denote the multivariate normal distribution with mean $\boldsymbol{\mu}$ and covariance matrix $\text{diag}(\boldsymbol{\sigma})$. If we rewrite the left-hand side of the PAC-Bayes bound by Taylor

expansion, we have:

$$
\begin{aligned}
\mathbb{E}_{\boldsymbol{\theta}\sim\mathcal{Q}_{\hat{\boldsymbol{\mu}},\hat{\sigma}}}\ell(f_{\boldsymbol{\theta}};\mathcal{D}) &= \mathbb{E}_{\Delta\boldsymbol{\theta}\sim\mathcal{Q}_{\mathbf{0},\hat{\sigma}}}\ell(f_{\hat{\boldsymbol{\theta}}+\Delta\boldsymbol{\theta}};\mathcal{D}) \\
&\approx \ell(f_{\hat{\boldsymbol{\theta}}},\mathcal{D}) + \mathbb{E}_{\Delta\boldsymbol{\theta}\sim\mathcal{Q}_{\mathbf{0},\hat{\sigma}}}(\nabla\ell(f_{\hat{\boldsymbol{\theta}}};\mathcal{D})^T\Delta\boldsymbol{\theta} + \frac{1}{2}\Delta\boldsymbol{\theta}^\top\nabla^2\ell(f_{\hat{\boldsymbol{\theta}}};\mathcal{D})\Delta\boldsymbol{\theta}) \\
&= \ell(f_{\hat{\boldsymbol{\theta}}};\mathcal{D}) + \frac{1}{2}\mathrm{Tr}(\mathrm{diag}(\hat{\boldsymbol{\sigma}})\nabla^2\ell(f_{\hat{\boldsymbol{\theta}}};\mathcal{D})) \geq \ell(f_{\hat{\boldsymbol{\theta}}};\mathcal{D}).
\end{aligned}
\tag{27}
$$

Recall here $\hat{\boldsymbol{\mu}}$ and $\hat{\boldsymbol{\sigma}}$ are the minimizers of the PAC-Bayes loss, obtained by solving the optimization problem Equation (P). Equation Equation (27) states that the deterministic predictor has a smaller prediction error than the Bayesian predictor. However, note that the last inequality in Equation (27) is derived under the assumption that the term $\nabla^2\ell(f_{\hat{\boldsymbol{\theta}}};\mathcal{D})$ is positive-semidefinite. This is a reasonable assumption as $\hat{\boldsymbol{\mu}}$ is the local minimizer of the PAC-Bayes loss, and the PAC-Bayes loss is close to the population loss when the number of samples is large. Nevertheless, since this property only approximately holds, the presented argument can only serve as an intuition that shows the potential benefits of using the deterministic predictor.

## C   Extended Experimental Details

We conducted experiments using eight A5000 GPUs with four AMD EPYC 7543 32-core Processors. To speed up the training process for posterior and prior variance, we utilized a warmup method that involved updating the noise level in the posterior of each layer as a scalar for the first 50 epochs and then proceeding with normal updates after the warmup period. This method only affects the convergence speed, not the generalization, and it was only used for large models in image classification.

### C.1   Parameter Settings

Recall that the exponential momentum bound $K(\boldsymbol{\lambda})$ is estimated over a range $[\gamma_1, \gamma_2]$ of $\gamma$ as per Definition 4.7. It means that we need the inequality

$$
\mathbb{E}_{\mathbf{h}\sim\mathcal{P}_{\boldsymbol{\lambda}}}\mathbb{E}[\exp\left(\gamma(\mathbb{E}[X(\mathbf{h})] - X(\mathbf{h}))\right)] \leq \exp\left(\gamma^2 K(\boldsymbol{\lambda})\right)
$$

to hold for any $\gamma$ in this range. One needs to be a little cautious when choosing the upper bound $\gamma_2$, because if it is too large, then the empirical estimate of $\mathbb{E}_{\mathbf{h}\sim\mathcal{P}_{\boldsymbol{\lambda}}}\mathbb{E}[\exp\left(\gamma(\mathbb{E}[X(\mathbf{h})] - X(\mathbf{h}))\right)]$ would have too large of a variance. Therefore, we recommended $\gamma_2$ to be set to no more than 10 or 20. The choice of $\gamma_1$ also does not seem to be very crucial, so we have fixed it to 0.5 throughout.

For large datasets (like in MNIST or CIFAR10), $m$ is large. Then, according to Theorem 4.15, we can set the range $M, T, a, b$ of the trainable parameters to be very large with only a little increase of the bound (as $M, T, a, b$ are inside the logarithm), and then during training, the parameters would not exceed these bounds even if we don't clip them. Hence, no clipping is needed for very large networks or with small networks with proper initializations. But when the dataset size $m$ is small, or the initialization is not good enough, then the correction term could be large, and clipping will be needed.

The clipping is also needed from the usual numerical stability point of view. As $\lambda$ is in the denominator of the KL-divergence, it cannot be too close to 0. Because of this, in the numerical experiments on GNN and CNN13/CNN15, we clip the domain of $\lambda$ at a lower bound of 0.1 and $5e-3$, respectively. For the VGG and Resnet experiments, the clipping $\lambda$ is optional.

### C.2   Baseline PAC-Bayes bounds for unbounded loss functions

We compared two baseline PAC-Bayes bounds when training CNNs with our layerwise PAC-Bayes bound. The bounds are expressed in our notation. Consider a neural network model denoted as $f_{\boldsymbol{\theta}}$, where $f$ represents the network's architecture, and $\boldsymbol{\theta}$ is the weight.

- sub-Gaussian (Theorem 5.1 of Alquier (2021)):

$$
\mathbb{E}_{\boldsymbol{\theta}\sim\mathcal{Q}}\ell(f_{\boldsymbol{\theta}};\mathcal{D}) \leq \mathbb{E}_{\boldsymbol{\theta}\sim\mathcal{Q}}\ell(f_{\boldsymbol{\theta}};\mathcal{S}) + \frac{1}{m\gamma}\left(\log\frac{1}{\delta} + \mathrm{KL}(\mathcal{Q}||\mathcal{P})\right) + K_{sub}\gamma,
\tag{28}
$$

where $K_{sub}$ is the variance factor by assuming the loss function $\ell$ is sub-Gaussian as defined below:

$$\mathbb{E}_{\boldsymbol{\theta} \sim \mathcal{P}} \mathbb{E}_{\mathcal{S} \sim \mathcal{D}} \exp\left[\gamma(\ell(f_{\boldsymbol{\theta}}; \mathcal{S}) - \ell(f_{\boldsymbol{\theta}}; \mathcal{D}))\right] \leq \exp\left(\gamma^2 K_{sub}\right), \forall \gamma \in \mathbb{R}.$$

In the experiment to generate Figure 1, we restricted $\gamma$ to $[-1, 1]$. Otherwise, if we use $\gamma \in \mathbb{R}$, the sub-Gaussian bound would be too large to calculate, resulting in a NaN value. Therefore, the reported $K_{sub}$ in Figure 1 may be underestimated.

- CGF (Theorem 9 of Rodríguez-Gálvez et al. (2023)):

$$\mathbb{E}_{\boldsymbol{\theta} \sim \mathcal{Q}} \ell(f_{\boldsymbol{\theta}}; \mathcal{D}) \leq \mathbb{E}_{\boldsymbol{\theta} \sim \mathcal{Q}} \ell(f_{\boldsymbol{\theta}}; \mathcal{S}) + \frac{1}{\gamma}\left(\frac{1}{m}(\log\frac{1}{\delta} + \mathrm{KL}(\mathcal{Q}||\mathcal{P})) + \psi(\gamma)\right), \tag{29}$$

where $\psi(\gamma)$ is a convex and continuously differentiable function defined on $[0, b)$ for some $b \in \mathbb{R}^+$ such that $\psi(0) = \psi'(0) = 0$ and $\mathbb{E}_{\boldsymbol{\theta} \sim \mathcal{P}} \mathbb{E}_{\mathcal{S} \sim \mathcal{D}}[\exp(\gamma(\ell(f_{\boldsymbol{\theta}}; \mathcal{D}) - \ell(f_{\boldsymbol{\theta}}; \mathcal{S})))] \leq \exp(\psi(\gamma))$ for all $\gamma \in [0, b)$. There is no specific form of $\psi(\gamma)$ provided in the original paper, but in order to achieve the normal $\sqrt{m}$ convergence rate of the PAC-Bayes bound, we need at least set $\psi(\gamma) = K_{CGF}\gamma^{\alpha}$ ($\alpha \geq 2$). Among these, using $\alpha = 2$ gives the smallest $K$, so we used $\alpha = 2$ for the comparison in Figure 1 and 3.

## C.3    Compatibility with Data Augmentation

We didn't include data augmentation in the experiments in the main text. Because with data augmentation, there is no rigorous way of choosing the sample size $m$ that appears in the PAC-Bayes bound. More specifically, for the PAC-Bayes bound to be valid, the training data has to be i.i.d. samples from some underlying distribution. However, most data augmentation techniques would break the i.i.d. assumption. As a result, if we have 10 times more samples after augmentation, the new information they bring in would be much less than those from 10 times i.i.d. samples. In this case, how to determine the effective sample size $m$ to be used in the PAC-Bayes bound is a problem.

Since knowing whether a training method can work well with data augmentation is important, we carried out the PAC-Bayes training with an ad-hoc choice of $m$, that is, we set $m$ to be the size of the augmented data. We compared the grid-search result of SGD and Adam versus PAC-Bayes training on CIFAR10 with ResNet18. The augmentation is achieved by random flipping and random cropping. The data augmentation increased the size of the training sample by 128 times. The test accuracy for SGD is 95.2%, it is 94.3% for Adam, it is 94.4% for AdamW, and it is 94.3% for PAC-Bayes training with the layerwise prior. In contrast, the test accuracy without data augmentation is lower than 90% for all methods. It suggests that data augmentation does not conflict with the PAC-Bayes training in practice.

## C.4    Model analysis

We examined the learning process of PAC-Bayes training by analyzing the posterior variance $\boldsymbol{\sigma}$ for different layers in models trained by Algorithm 3. Typically, batch norm layers have smaller $\boldsymbol{\sigma}$ values than convolution layers. Additionally, shadow convolution and the last few layers have smaller $\boldsymbol{\sigma}$ values than the middle layers. We also found that skip-connections in ResNet18 have smaller $\boldsymbol{\sigma}$ values than nearby layers, suggesting that important layers with a greater impact on the output have smaller $\boldsymbol{\sigma}$ values.

In Stage 1, the training loss is higher than the testing loss, which means the adopted PAC-Bayes bound is able to bound the generalization error throughout the PAC-Bayes training stage. Additionally, we observed that the final value of $K$ is usually very close to the minimum of the sampled function values. The average value of $\boldsymbol{\sigma}$ experienced a rapid update during the initial 50 warmup epochs but later progressed slowly until Stage 2. The details can be found in Figure 9 and 10. Based on the figures, shadow convolution, and the last few layers have smaller $\boldsymbol{\sigma}$ values than the middle layers for all models. We also found that skip-connections in ResNet18 and ResNet34 have smaller $\boldsymbol{\sigma}$ values than nearby layers on both datasets, suggesting that important layers with a greater impact on the output have smaller $\boldsymbol{\sigma}$ values.

**Computational cost**: In PAC-Bayes training, we have four parameters $\boldsymbol{\mu}, \boldsymbol{\lambda}, \boldsymbol{\sigma}, \gamma$. Among these variables, $\gamma$ can be computed on the fly or whenever needed, so there is no need to store them. We need to store $\boldsymbol{\mu}, \boldsymbol{\lambda}, \boldsymbol{\sigma}$,

where $\sigma$ has the same size as $\mu$ and the size of $\lambda$ is the same as the number of layers which is much smaller. Hence the total storage is approximately doubled. Likewise, when computing the gradient for $\mu, \lambda, \sigma$, the cost of automatic differentiation in each iteration is also approximately doubled. In the inference stage, the complexity is the same as in conventional training.

**Effect of two stages**: We have tested the effect of the two stages. Without the first stage, the algorithm cannot automatically learn the noise level and weight decay to be used in the second stage. If the first stage is there but too short (10 epochs for example), then the final performance of VGG13 on CIFAR100 will reduce to 64.0% . Without Stage 2, the final performance is not as good as reported either. The test accuracy of models like VGG13 and ResNet18 on CIFAR10 would be 10% lower as in Figure 9 and 10.

### C.5 Node classification by GNNs

We test the PAC-Bayes training algorithm on the following popular GNN models, tuning the learning rate $(1e{-}3, 5e{-}3, 1e{-}2)$, weight decay $(0, 1e{-}4, 1e{-}3, 1e{-}2)$, noise injection $(0, 1e{-}3, 5e{-}3, 1e{-}2)$, and dropout $(0, 0.4, 0.8)$. The number of filters per layer is 32 in GCN (Kipf & Welling, 2016) and SAGE (Hamilton et al., 2017). For GAT (Veličković et al., 2017), the number of filters is 8 per layer, the number of heads is 8, and the dropout rate of the attention coefficient is 0.6. Fpr APPNP (Gasteiger et al., 2018), the number of filters is 32, $K = 10$ and $\alpha = 0.1$. We set the number of layers to 2, achieving the best baseline performance. A ReLU activation and a dropout layer are added between the convolution layers for baseline training only. Since GNNs are faster to train than convolutional neural networks, we tested all possible combinations of the above parameters for the baseline, conducting 144 searches per model on one dataset. We use Adam as the optimizer with the learning rate as $1e{-}2$ for all models using both training and validation nodes for PAC-Bayes training.

We also did a separate experiment using both training and validation nodes for training. For baselines, we need first to train the model to detect the best hyperparameters as before and then train the model again on the combined data. Our PAC-Bayes training can also match the best generalization of baselines in this setting.

All results are visualized in Figure 5-8. The AdamW+val and scalar+val record the performances of the baseline and the PAC-Bayes training, respectively, with both training and validation datasets for training. We can see that test accuracy after adding validation nodes increased significantly for both methods but still, the results of our algorithm match the best test accuracy of baselines. Our proposed PAC-Bayes training with the scalar prior is better than most of the settings during searching and achieved comparable test accuracy when adding validation nodes to training.

### C.6 Few-shot text classification with transformers

The proposed method is also observed to work on transformer networks. We conducted experiments on two text classification tasks of the GLUE benchmark as shown in Table 6. SST is the sentiment analysis task, whose performance is evaluated as the classification accuracy. Sentiment analysis is the process of analyzing the sentiment of a given text to determine if the emotional tone of the text is positive, negative, or neutral. QNLI (Question-answering Natural Language Inference) focuses on determining the logical relationship between a given question and a corresponding sentence. The objective of QNLI is to determine whether the sentence contradicts, entails, or is neutral with respect to the question.

We use classification accuracy as the evaluation metric. The baseline method uses grid search over the hyper-parameter choices of the learning rate $(1e{-}1, 1e{-}2, 1e{-}3)$, batch size $(2, 8, 16, 32, 80)$, dropout ratio $(0, 0.5)$, optimization algorithms (SGD, AdamW), noise injection $(0, 1e{-}5, 1e{-}4, 1e{-}3, 1e{-}2, 1e{-}1)$, and weight decay $(0, 1e{-}1, 1e{-}2, 1e{-}3, 1e{-}4)$. The learning rate and batch size of our method are set to $1e{-}3$ and 100 (i.e., full-batch), respectively. In this task, the number of training samples is small (80). As a result, the preset $\gamma_2 = 10$ is a bit large and thus prevents the model from achieving the best performance with PAC-Bayes training.

We adopt BERT (Devlin et al., 2018) as our backbone and added one fully connected layer as the classification layer. Only the added classification layer is trainable, and the pre-trained model is frozen without gradient

update. To simulate a few-shot learning scenario, we randomly sample 100 instances from the original training set and take the whole development set to evaluate the classification performance. We split the training set into 5 splits, taking one split as the validation data and the rest as the training set. Each experiment was conducted five times, and we report the average performance. We used the PAC-Bayes training with the scalar prior in this experiment. According to Table 6, our method is competitive to the baseline method on the SST task, and the performance gap is only 0.4 points. On the QNLI task, our method outperforms the baseline by a large margin, and the variance of our proposed method is less than that of the baseline method.

Table 6: Test accuracy on the development sets of 2 GLUE benchmarks.

|  | SST | QNLI |
|---|---|---|
| baseline | **72.9±0.99** | 62.6±0.10 |
| scalar | 72.5±0.99 | **64.2±0.02** |

### C.7 Additional experiments stability

We conducted extra experiments to showcase the robustness of the proposed PAC-Bayes training algorithm. Specifically, we tested the effect of different learning rates on ResNet18 and VGG13 models trained with layerwise prior. Learning rate has long been known as an important impact factor of the generalization for baseline training. Within the stability range of gradient descent, the larger the learning rate is, the better the generalization has been observed (Lewkowycz et al., 2020). In contrast, the generalization of the PAC-Bayes trained model is less sensitive to the learning rate. We do observe that due to the newly introduced noise parameters, the stability of the optimization gets worse, which in turn requires a lower learning rate to achieve stable training. But as long as the stability is guaranteed by setting the learning rate low enough, our results, as Table 7, indicated that the test accuracy remained stable across various learning rates for VGG13 and Resnet18. The dash in the table means that the learning rate for that particular setting is too large to maintain the training stability. For learning rates below $1e-4$, we trained the model in Stage 1 for more epochs (700) to fully update the prior and posterior variance.

We also demonstrate that the warmup iterations (as discussed at the beginning of this section) do not affect generalization. As shown in Table 9, the test accuracy is insensitive to different numbers of warmup iterations. Furthermore, additional evaluations of the effects of batch size (Table 10), optimizer (Tables 11), and $\gamma_1$ and $\gamma_2$ (Table 12)

Table 7: Test accuracy of ResNet18 trained with different learning rates.

| lr | $3e-5$ | $5e-5$ | $1e-4$ | $2e-4$ | $3e-4$ | $5e-4$ |
|---|---|---|---|---|---|---|
| CIFAR10 | 88.4 | 88.8 | 89.3 | 88.6 | 88.3 | 89.2 |
| CIFAR100 | 69.2 | 69.0 | 68.9 | 69.1 | 69.1 | 69.6 |

Table 8: Test accuracy of VGG13 trained with different learning rates.

| lr | $3e-5$ | $5e-5$ | $1e-4$ | $2e-4$ | $3e-4$ | $5e-4$ |
|---|---|---|---|---|---|---|
| CIFAR10 | 88.6 | 88.9 | 89.7 | 89.6 | 89.6 | 89.5 |
| CIFAR100 | 67.7 | 68.0 | 67.1 | - | - | - |

Table 9: Test accuracy of ResNet18 trained with warmup epochs of $\sigma$.

|          | 10   | 20   | 50   | 80   | 100  | 150  |
|----------|------|------|------|------|------|------|
| CIFAR10  | 88.5 | 88.5 | 89.3 | 89.5 | 89.5 | 88.9 |
| CIFAR100 | 69.4 | 69.6 | 68.9 | 69.1 | 69.0 | 68.1 |

Table 10: Test accuracy of VGG13 with different batch sizes.

| Batch Size | 128  | 256  | 1024 | 2048 | 2500 |
|------------|------|------|------|------|------|
| Test Acc   | 89.7 | 89.7 | 88.7 | 89.4 | 88.3 |

Table 11: Test accuracy of ResNet18 using SGD: Effects of different momentum values (with learning rate $1 \times 10^{-3}$) and different learning rates (with momentum 0.9).

|          | Momentum | | | Learning Rate | | |
|----------|------|------|------|--------------------|--------------------|--------------------|
|          | 0.3  | 0.6  | 0.9  | $1 \times 10^{-4}$ | $3 \times 10^{-4}$ | $1 \times 10^{-3}$ |
| Test Acc | 88.6 | 88.8 | 89.2 | 88.3 | 88.8 | 89.2 |

Table 12: Test accuracy of ResNet18 with different settings for $\gamma_1$ (with $\gamma_2 = 20$) and $\gamma_2$ (with $\gamma_1 = 0.1$).

|          | $\gamma_1$ | | | $\gamma_2$ | | |
|----------|------|------|------|------|------|------|
|          | 0.1  | 0.5  | 1.0  | 10   | 15   | 20   |
| Test Acc | 88.8 | 89.3 | 88.8 | 89.3 | 89.4 | 89.4 |

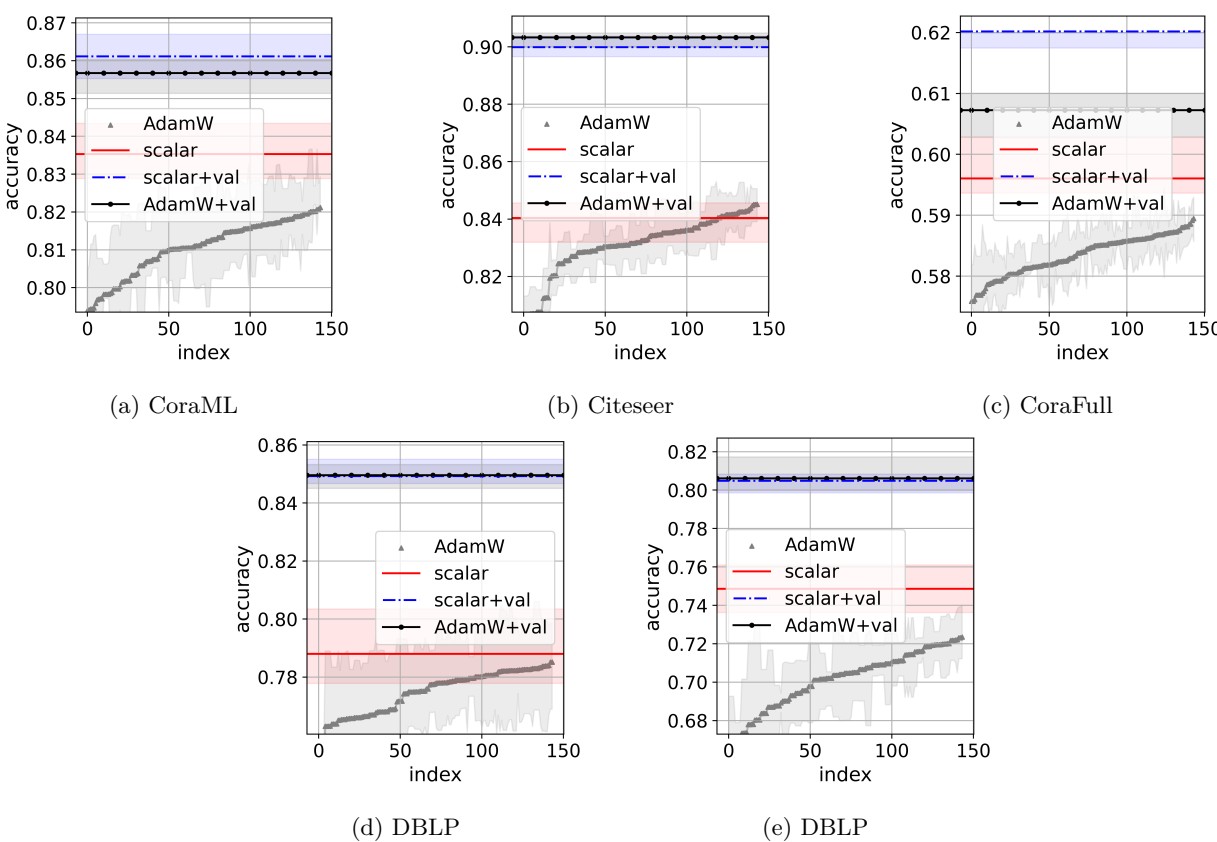

(a) CoraML  (b) Citeseer  (c) CoraFull

(d) DBLP  (e) DBLP

Figure 5: Test accuracy of GCN. The first and third quartiles construct the interval over the ten random splits. {+val} denotes the performance with both training and validation datasets for training.

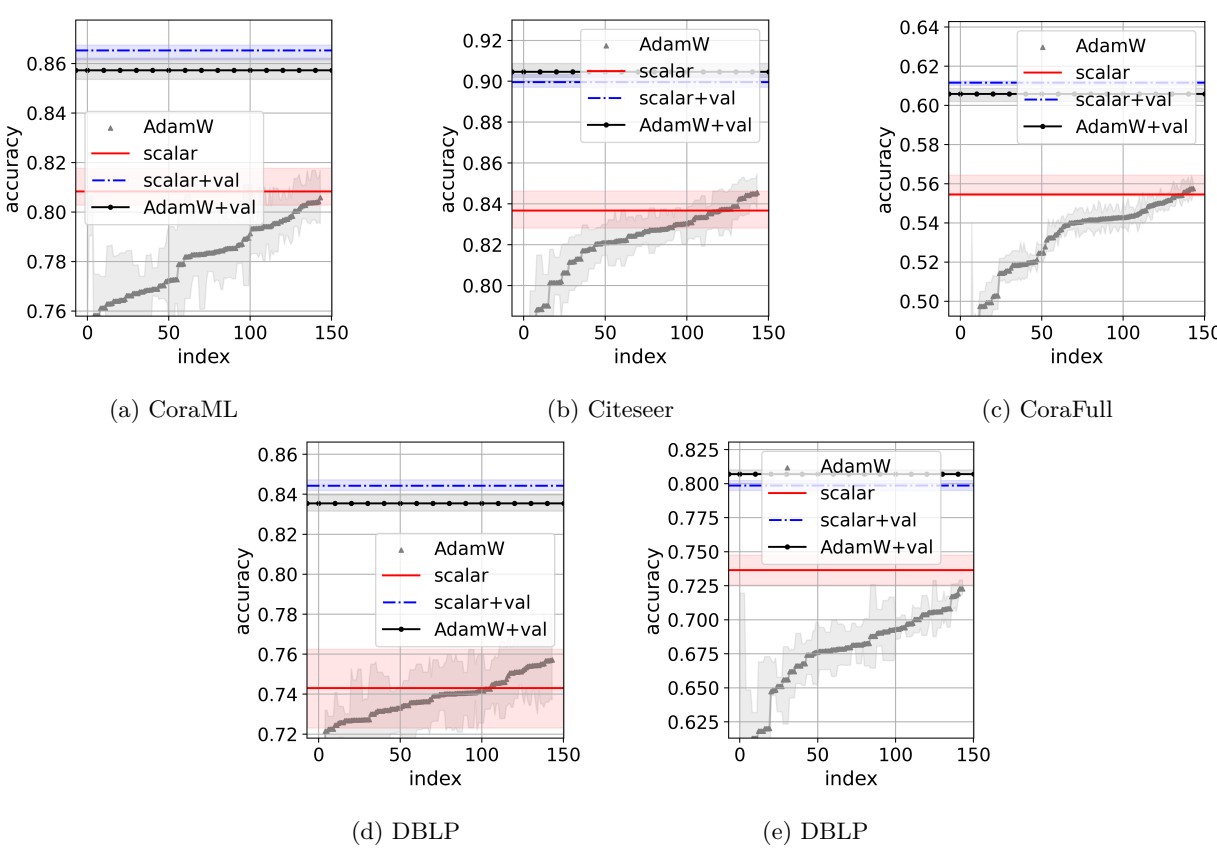

Figure 6: Test accuracy of SAGE. The first and third quartiles construct the interval over the ten random splits. {+val} denotes the performance with both training and validation datasets for training.

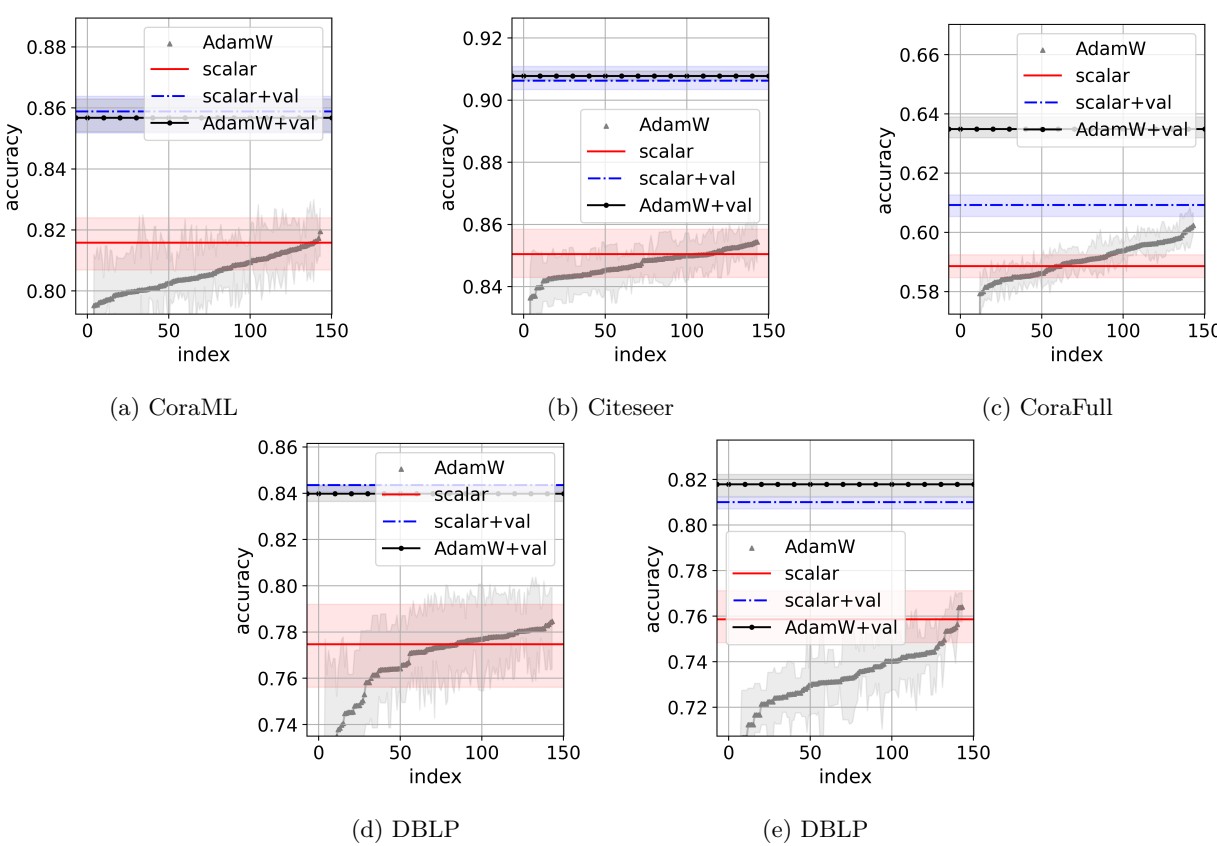

(a) CoraML          (b) Citeseer          (c) CoraFull

(d) DBLP          (e) DBLP

Figure 7: Test accuracy of GAT. The first and third quartiles construct the interval over the ten random splits. {+val} denotes the performance with both training and validation datasets for training.

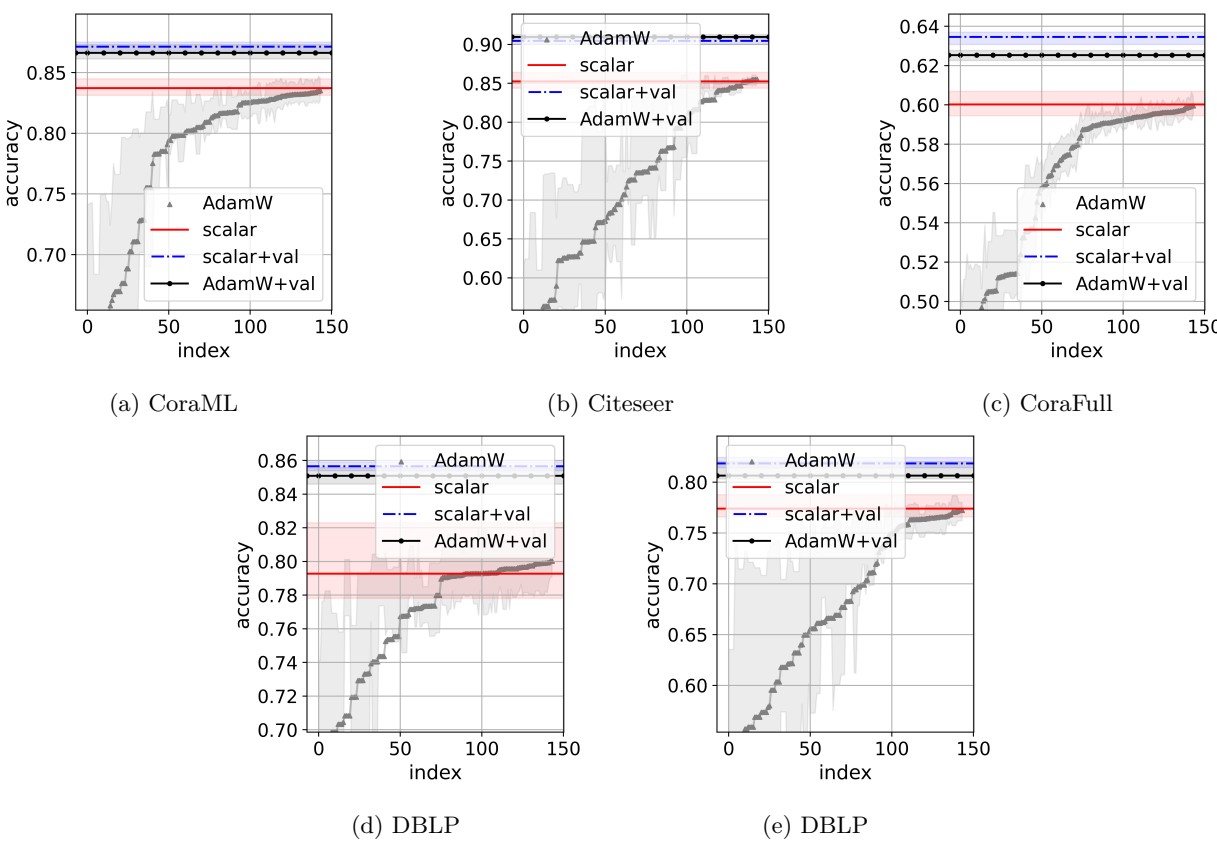

Figure 8: Test accuracy of APPNP. The first and third quartiles construct the interval over the ten random splits. {+val} denotes the performance with both training and validation datasets for training.

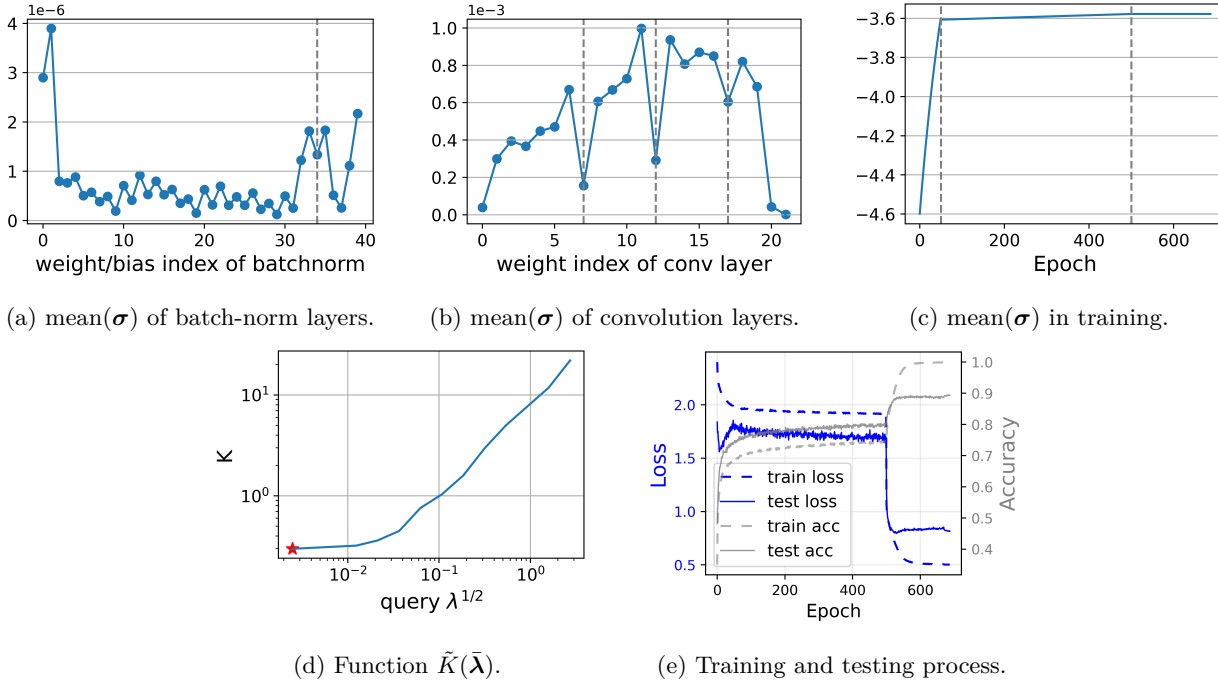

Figure 9: Training details of ResNet18 on CIFAR10. The red star denotes the final $K$.

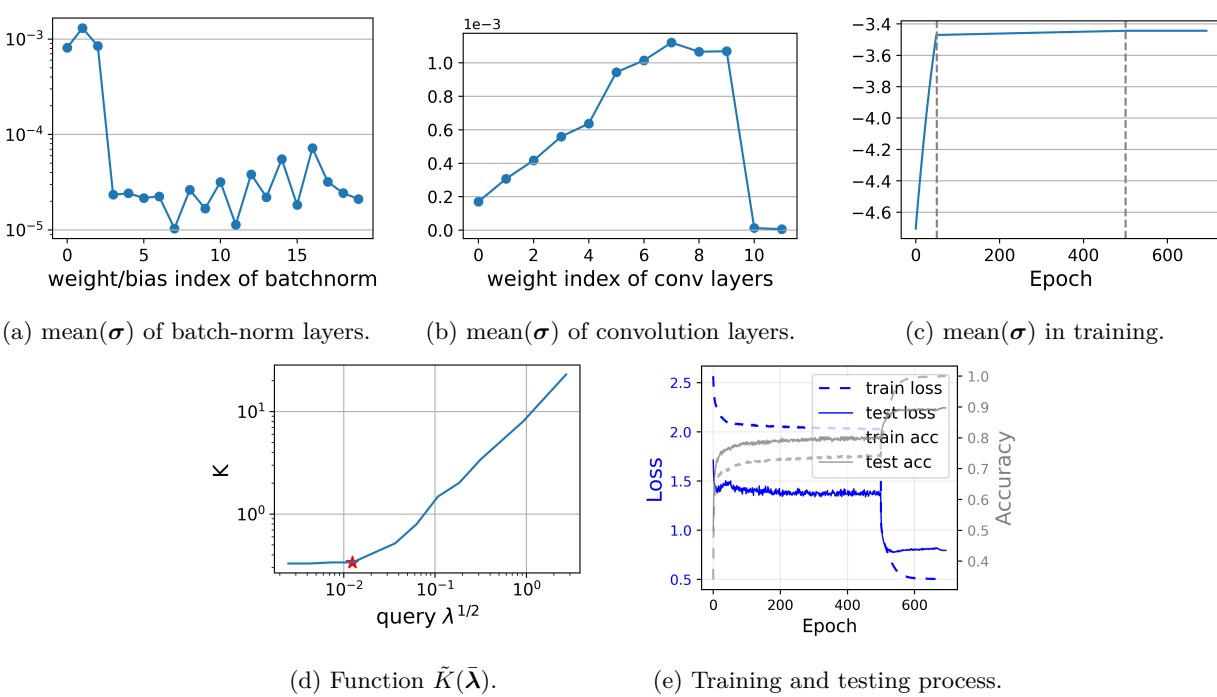

(a) mean($\boldsymbol{\sigma}$) of batch-norm layers.

(b) mean($\boldsymbol{\sigma}$) of convolution layers.

(c) mean($\boldsymbol{\sigma}$) in training.

(d) Function $\tilde{K}(\bar{\boldsymbol{\lambda}})$.

(e) Training and testing process.

Figure 10: Training details of VGG13 on CIFAR10. The red star denotes the final $K$.

