# OpenReview forum: "Improving Generalization of Complex Models under Unbounded Loss Using PAC-Bayes Bounds"
_TMLR — Accepted by TMLR_

### Review · Reviewer_g1cP · 2024-05-09

**Summary Of Contributions:**

Authors introduce a new PAC-Bayes training algorithm with improved performance and reduced reliance on prior tuning by proposing a new PAC-Bayes bound for unbounded loss and a theoretically grounded approach that involves jointly training the prior and posterior using the same dataset. Results agains both PAC and ERM based model support the proposal.

**Audience:**

Yes

**Claims And Evidence:**

Yes

**Requested Changes:**

Authors should stress more the novelty of the proposal.
Authors should better clarify the selection of the experimental results.

**Strengths And Weaknesses:**

Strengths
- new theoretical results
- good empirical results
Weaknesses
- authors should stress more the novelty of the proposal
- authors should better clarify the selection of the experimental results

---

> ### Author Response · Authors · 2024-06-25
>
> Thank you very much for the feedback! We have revised our manuscript to address your concerns:
>
> 1) We added a motivation section (Sec. 4.1) to illustrate why the previous PAC-Bayes bounds are insufficient, thus motivating our work.
> 2) We modified and shortened the contribution statement (page 2) to highlight only the main contribution.
> 3) We added Remark 4.12 and Figures 1 and 2 to explain where the improvement of our new bound comes from and how much tighter it is numerically compared to existing bounds.
> 4) We modified the titles of sections and subsections to make the structure clearer.
>
> All changes are highlighted. Please let us know if further clarification is needed.

---

### Review · Reviewer_4s6g · 2024-05-29

**Summary Of Contributions:**

To start, I will briefly summarize by understanding of the PAC-Bayes literature which acts as a broad context for this paper. Typical PAC-Bayes bounds for machine learning tasks are formulated for "randomized" learning algorithms, i.e., we have a hypothesis class (a set of model candidates, e.g., a set of classifiers) over which we consider so-called "prior" and "posterior" distributions; the candidate selected by the posterior distribution is the "output" of a randomized learning algorithm. As in standard statistical learning theory, we have a loss function depending on a (model, data) pair, and the ultimate objective of interest is the expected loss incurred by the aforementioned learning algorithm output. Here "expected loss" includes taking expectation with respect to a new test point as well as expectation over the posterior; this is sometimes referred to as the "Gibbs risk" if I recall correctly. One important quality of PAC-Bayes bounds is that they hold with high probability (over training data), uniform in the choice of posterior distribution, so the posterior is allowed to depend on training data.

The notion of "PAC-Bayes training" plays a central role in the paper being reviewed; as just mentioned, the choice of posterior can depend on training data, and thus in principle it can be "optimized" based on various empirical criteria. Indeed, much of the past PAC-Bayes literature provides (bound, algorithm) pairs as their main results. Under various assumptions new bounds are derived that are completely (or partially) determined by data available at training time, and algorithms are derived as procedures that minmize those bounds in terms of the posterior (either exactly or approximately).

From what I gather, the main interest of the authors, in terms of models, is neural networks (for classification). Their main motivation overall is to design PAC-Bayes training algorithms which (without much or any tuning) perform just as well as traditional empirical risk minimization (ERM) with lots of tuning. Of course, plenty of past research has looked at neural network PAC-Bayes training with the same goals. The main technical points which the authors emphasize differentiate this work from existing literature are as follows: losses are allowed to be unbounded, bounds to be optimized are completely empirical, and the *prior* distribution can also be data-dependent without having to split the data into two disjoint subsets. In addition, they provide a practical training algorithm assuming the distribution over the hypothesis class is Gaussian, and empirically evaluate their procedure, which they find to be competitive.

Their main result in Thm 4.9 is obtained by assuming a Lipschitz property on the KL divergence and the prior-dependent exponential moment bound.

Their proposed algorithm is a "two-stage" procedure which is not purely PAC-Bayes training, but rather does typical PAC-Bayes training on their objective, and then fixes everything but the mean of the posterior and does essentially ERM on the Gibbs empirical risk. My overall impression is that this practical modification is what leads their proposal to get closer to well-tuned ERM performance.

**Audience:**

Yes

**Broader Impact Concerns:**

Not applicable.

**Claims And Evidence:**

No

**Requested Changes:**

Here I will highlight some points I found confusing or troublesome. I'm not worried about novelty, but the exposition and main claims have some rather significant issues of clarity in my opinion.


__Exposition of the exponential moment approach__

A lot of space is dedicated to the exponential moment bound approach starting in Defn 4.1 (and going through Lemma 4.2 to Defn 4.4), but I found the overall logic a bit hard to parse, and I feel like other readers would as well. The authors claim that their exponential moment condition is "weaker than the 2nd-order moment condition" (page 4). I get that *if* we only need one-sided bounds (i.e., for $\\mathbb{E}[X]-X$ but not for $X-\\mathbb{E}[X]$) and $X \\geq 0$, then the exponential bound is weaker than the second moment bound, assuming $\\gamma$ is small enough. That said, for newcomers to PAC-Bayes who read "Unbounded Loss" in the title, one expects they would assume that control of $X-\\mathbf{E}[X]$ will be important, i.e., the right-hand tails of the loss distribution will cause problems for analysis. If this is not the case, and a one-sided approach is sufficient (even if say the right tails are heavy), some exposition of *why* would be meaningful.

Next, the authors also state that the main motivation for Defn 4.1 is to get "a bound with a smaller numerical value." This is unclear. A smaller numerical value when compared with *what?* In the proof of Lemma 4.2, the authors mention how setting $K$ to the larger of $\\text{Var}(X)$ or $(\\mathbb{E}X)^{2}$ is sufficient for the condition in Defn 4.1 to hold; are these values supposed to be compared with $\\mathbb{E}X^{2}$, which is no smaller than either of the previous two quantities?


__Claims regarding empirical bounds__

In Remark 4.6, the authors emphasize that their bound *"seems to be the first purely empirical bound (i.e., computable from data) that can be easily used for training."* I take this is to be a key claim for this paper, as it corresponds as a direct answer to the limitation highlighted in previous literature in the last sentence of the first paragraph of section 3 (*"... all suffer from this issue."*). The latter part of this claim ("can easily be used for training") is unclear, but more importantly, upon what do the authors base their claim that the bound in Thm 4.5 is "purely empirical"? As I read it from Defn 4.4, the key new quantity $K(\\boldsymbol{\\lambda})$ depends critically on the population distribution. Maybe I'm missing something, but this is a key point that needs clarification.


__Unclear main claims__

Considering the [acceptance criteria](https://jmlr.org/tmlr/acceptance-criteria.html) for TMLR, the most important point is that submissions need to have claims that are supported by accurate, convincing, and clear evidence. For evidence to be meaningful, the claims that the evidence supports must of course be clear, but I feel this point is lacking in the current submission. I'll try and be concise by placing my focus on the summary of contributions at the end of section 1. Some of these points are really vague and not well-rooted in the substantive content of sections 4-6. A bit more detail:

- Point 1 (*"We introduce a new..."*): Yes, the authors do introduce a new bound, but how does it compare to previous bounds? In sections 3-4 one gets the impression that smaller bounds that are purely empirical is a key point, but this isn't touched on in the contribution summary. Of course, the evidence is also unclear, as I pointed out earlier. The second sentence in point 1 (*"This algorithm simultaneously optimizes..."*) is very misleading. If the authors are referring to the algorithm in section 5, the second stage is in my opinion a critical point, yet it is glossed over, making it seem like their practical algorithm is just a pure PAC-Bayes bound minimizer.

- Point 2 (*"The test performance of the proposed algorithm is theoretically justified."*"): this is a totally ethereal statement, really impossible to prove or disprove, and should not be positioned as a main claim. Furthermore, even for the casual reader, intuitively, since stage 2 of Algorithm 1 is a big heuristic modification, it doesn't really jive with this vague claim.

- Points 3, 4, 5: these points are quite lengthy, and really should be summarized into a single point that discusses the actual proposed algorithm and its strengths/weaknesses found in the empirical tests.


__Technical points can be quite sloppy__

There are numerous points I found troublesome related to notation and technical exposition. Here is a handful of representative examples.

- Page 5: parameterizing the posterior distribution, the authors immediately dive into parameterization with mean and variance vectors $\\mathbf{h}$ and $\\boldsymbol{\\sigma}$, but this is quite unnatural. At this stage in the paper, the set of distributions from which the posterior is taken need not be Gaussian, yet the formulation already essentially assumes the Gaussian special case that comes up in section 5. I personally feel this kind of inconsistency in generality can spoil the exposition. The content in section 4 in no way depends on the distribution family being Gaussian.

- Page 5 (and all over the place): the poor usage of $\\mathbf{h}$ and $\\theta$ really clutters the notation and makes things inconsistent. For example, in section 2, $\\mathbf{h}$ is a classifier which returns a *"predicted label"*. This is consistent with the text before Defn 4.4, where $\\mathbf{h} = f\_{\\theta}$, but does not match up with the last paragraph of page 5, where $\\mathbf{h} \\in \\mathbb{R}^{d}$ and $\\mathbf{h}$ has been re-appropriated to do what really should be done by $\\theta$ parameterizing the family $\\{f\_{\\theta}: \\theta \\in \\mathbb{R}^{d}\\}$. The authors make this switch in section 5, but this really should be cleaned up a bit.

- Thm 4.9: the bound holds *"with probability at least $1-\\epsilon$"*, and yet $\\epsilon$ appears nowhere. Presumably it is buried within $L\_{PAC}$, whose original definition uses $\\delta$ rather than $\\epsilon$.

- Thm 4.9: $C$ is said to be a "constant" but it can also depend on $\\eta\_{2}$; this hides the true order of $\\eta_{2}(\\varepsilon)$ in the bound (3).


__Miscellaneous points__

- $\\mathrm{P}$ that appears in the probability space notation in Defn 4.1 looks identical to equation (P) on page 5. The former isn't really critical, is it?
- Double spaces (two consecutive spaces between words): e.g., Corollary 5.1 third bullet.
- Most of the ideas underlying this paper are totally model agnostic, i.e., they hold regardless of what $\\mathcal{H}$ is, and yet for some reason the authors start in the *abstract* by talking about updating *"the network weights"* without any mention of neural networks before that. This is another example of inconsistency in terms of generality, which I mentioned earlier regarding the posterior parameterization.

**Strengths And Weaknesses:**

__Strengths:__

I would say the strongest point of this paper is that the authors have developed a practical procedure that seems to work quite well on a variety of datasets and neural network architectures. Their basic motivations for designing this procedure are explained clearly, and rooted (partially) in solid theory. In addition, overall, the paper is quite well written, easy to read, and does a pretty good job of describing previous research and its relation to the present work.


__Weaknesses:__

On the other hand, the presentation of some technical points is rather sloppy, and most importantly (in terms of review standards at TMLR) I feel like the main claims and relevant evidence in this work are not sufficiently clear, leaving readers wondering "what have I learned?" after reading this paper. Yes, empirical tests suggest that the authors proposed procedure is "good" when our definition of "good" is "behaves like a well-tuned ERM algorithm", but how this relates to the novel content in section 4 is not in my opinion clear at all, and the main claims stated explicitly by the authors are not well-organized, and scattered throughout the paper. I will address these points in a bit more detail in the next section.

---

> ### Author Response · Authors · 2024-06-25
>
> We sincerely thank the reviewer for the constructive suggestions. Based on these recommendations, we have carefully revised our manuscript. Here are our responses to the requested changes.
>
> **Exposition of the exponential moment approach**
>
> >**1.  Revise the logic of Defn 4.1 (and go through Lemma 4.2 to Defn 4.4).**
>
> Thank you for your suggestion! Upon revision, Lemma 4.2 has become Lemma 4.5. Similarly, Def. 4.4 is now Def. 4.7 in the revised version.
>
> The advantage of Definition 4.1 becomes fully apparent only after the PAC-Bayes bound is introduced. However, we strive to provide some preliminary context to help readers better understand Definition 4.1. To achieve this, we have revised our extensive comments on Def. 4.1 into several remarks for better clarity. These remarks now include:
>
>  1). Discussions on the relationship between Def. 4.1 and previous work. (Remarks 4.3, 4.4).
>
>  2). Guidance for newcomers on understanding Def. 4.1. (Remarks 4.2) .
>
> Please refer to pages 4 and 5 for these revisions. We hope the content is now conveyed more effectively.
>
> Def. 4.7 is a necessary extension of Def. 4.1 for use in the PAC-Bayes bound. In the revised version, we have added Remark 4.8 and a paragraph after Remark 4.8 to clarify the difference between this definition and those in the literature.
>
> We also added Remark 4.12 and Figure 1 to illustrate how Def 4.1/4.7 leads to a smaller numerical value of the new PAC-Bayes bound.
>
> In short, the two key aspects of our $K$ design are: first, making it dependent on the limited range $\gamma \in [\gamma_1, \gamma_2]$, and second, making it dependent on the prior parameter. In the revised submission, we also highlighted the importance of $K$ in Remark 4.12. Our major contribution to tightening the bound lies in the design of \( K \) and showing the numerical comparison of different forms of $K$.
>
> >**2. Comments on the one-sided bound.**
>
> Thank you for the good suggestion. We have added a discussion in Remark 4.2 about why the one-sided approach is sufficient. Our definition includes $\mathbb{E}X - X$ in the exponent because it leads to an upper bound for the population loss when $\gamma>0$, which is what we need. Conversely, having $X-\mathbb{E}X$ in the exponent with positive $\gamma$ would result in a lower bound for the population loss.
>
> >**3. "a bound with a smaller numerical value" is unclear; why not compare with $\mathbb{E}X^2$ based on the proof of Lemma 4.2.**
>
> Def. 4.4 is now Def. 4.7 in the revised version.
>
> Sorry for the confusion. We want to "get a smaller numerical value" compared with other baselines PAC-Bayes bounds for unbounded loss.
>
> To clarify this point, we made the following modifications to the manuscript.
>
> We added a motivation section, Sec. 4.1, to discuss issues with existing bounds in greater detail to motivate our new bound. We added Remark 4.12 to explain why the new condition in Def. 4.1/4.7 leads to a smaller PAC-Bayes bound by introducing a smaller numerical value for $K$. We justified the "smaller numerical value" in Figure 1 by comparing it with the $K$ values in previous PAC-Bayes bounds for unbounded loss.
>
> Please note that we didn't use the upper bound  $\max (K_1, K_2)$ as our $K$ because it is still too large; instead, we estimate $K$ from data as discussed at the beginning of Sec. 4.5. Lastly, we added Figure 2 in Sec. 4.5, where we show how large the upper bound  $\max (K_1, K_2)$  could be, which motivates our proposed algorithm for estimating $K$.
>
> **Claims regarding empirical bounds**
> >**1. In Remark 4.6, upon what do the authors base their claim that the bound in Thm 4.5 is "purely empirical"?**
>
> Thank you for the question. We agree that this point needs clarification.
>
> We now provide an explanation in the third paragraph of Sec. 4.1 regarding why the existing bound with the second-order moment condition is semi-empirical and difficult to estimate from data. Essentially, this bound includes a population second-order moment of the loss **with respect to the posterior distribution**, which can be hard to estimate.
>
> For our $K$, even though it also involves the estimation of an expectation, the expectation is with respect to the prior distribution. Since the prior distribution we use to compute $K$ is independent of the data, the data are still i.i.d. when conditioned on the prior. This allows us to use the empirical mean to approximate the population mean. This contrasts with the previous bound under the second-order moment condition, where the expectation is with respect to the posterior distribution, making the data no longer i.i.d. when conditioned on the posterior.
>
> **Unclear main claims**
>
> We understand your point and have revised the main contribution as suggested. In short, now we only claim two contributions, both well supported by evidence in the paper: a new PAC-Bayes bound for unbounded loss that is numerically tighter than previous bounds for unbounded loss; 2. the addition of the second stage.

---

> ### Author Response · Authors · 2024-06-25
>
> **Revise technical points.**
> >**1. Notations of prior and posterior distributions.**
>
>  Thank you for pointing this out. We have removed the use of mean and variance before parameterizing the posterior and prior with Gaussian distributions. Before discussing the Gaussian case, we use $\mathbf{w}$ and $\boldsymbol{\lambda}$ to denote the parameters of the posterior and prior. Later, the Gaussian posterior and prior are denoted as $Q_{\boldsymbol{\mu}, \boldsymbol{\sigma}}, P_{\boldsymbol{\mu}_0, \boldsymbol{\lambda}}$,
> where $\boldsymbol{\mu}$ is the current mean and $\boldsymbol{\mu}_0$ is the random initialization. $\boldsymbol{\sigma}$ and $\boldsymbol{\lambda}$ denote the covariance parameters in the posterior and prior, respectively.
>
> >**2. Notations of the model and neural network.**
>
> We have removed the use of $\mathbf{h}$ to denote both the model and model parameters in different contexts. We chose other symbols to denote the weights and retained $\mathbf{h}$ to denote the model.
>
> >**3. Probability $\delta$ and $\epsilon$.**
>
> Upon revision, Thm 4.5 has become Thm 4.9. Similarly, Thm 4.9 is now Thm 4.15 in the revised version.
>
> Sorry for the confusion. $\epsilon$ denotes the probability of failure for Thm 4.15, similar to the $\delta$ for Thm 4.9, and their relation is $\epsilon:= \left(n(\varepsilon) + \frac{\gamma_2 - \gamma_1}{\varepsilon}\right)\delta$. We have also added this to the statement of Thm 4.15 in red.
>
> >**4.Thm 4.9 (Thm 15 now): $C$ is said to be a "constant" but it can also depend on $\eta_2$.**
>
> Sorry for the confusion. Our previous notation $C(\eta_1+\eta_2)$ does not mean $C$ is a function of $\eta_1$ and $\eta_2$, it means the constant $C$ times $\eta_1+\eta_2$. To avoid confusion, we have added a dot between $C$ and $(\eta_1+\eta_2)$.
>
> **Miscellaneous points**
>
> >**1. $\mathrm{P}$ that appears in the probability space notation in Defn 4.1 looks identical to equation (P) on page 5.**
>
> We have replaced the first $\mathrm{P}$ by $\mathcal{P}$.
>
> > **2. Double spaces.**
>
>  We have fixed the double spaces in the current version.
>
> > **3. Most of the ideas underlying this paper are totally model agnostic.  However, the authors start in the abstract by talking about updating "the network weights" without any mention of neural networks before that.**
>
> Yes, we agree that the ideas are mostly model agnostic. To address this concern, we have updated the abstract and added a paragraph at the end of Section 2 to discuss the reasons for focusing on neural networks in this work.

---

### Review · Reviewer_Gcra · 2024-07-08

**Summary Of Contributions:**

The authors present a new PAC-Bayes algorithm with theoretical analysis. They consider unbounded loss functions with a relaxed exponential moment assumption for hypotheses. Additionally, they propose an optional second stage of Bayesian training to further increase training accuracy. They conduct experiments on several datasets using CNNs and GNNs and compare the results with other baselines.

**Audience:**

Yes

**Broader Impact Concerns:**

N. A.

**Claims And Evidence:**

Yes

**Requested Changes:**

The new algorithm is compared with other selected PAC-Bayes baselines. It would be better to provide some additional comparisons using the evaluation metrics considered by these baselines.

**Strengths And Weaknesses:**

Strengths:
1. The authors provide a new PAC-Bayes algorithm with a relaxed assumption.
2. The new algorithm outperforms other selected PAC-Bayes baselines with mean predictors and achieves results comparable to the best in test accuracy.

Weaknesses:
It is interesting to explore the PAC-Bayes training objectives. If only the test accuracy is evaluated, it is unclear why this new algorithm, which considers both prior and posterior training and involves more hyperparameters than ERM, is preferable. It would be better to show how accuracy changes over time or to provide additional evaluation metrics.

---

> ### Author Response · Authors · 2024-07-11
>
> Thank you for finding our approach interesting!
>
> We showed, in terms of the final performance, that our PAC-Bayes training is better than other PAC-Bayes algorithms, and is comparable with (similar but not noticeably better) than ERM. But in terms of stability to hyperparameters, the PAC-Bayes training is much better.
>
> We agree that showing not only the final accuracy but also the accuracy during training would be informative. Hence we added Figure 4 to the revised manuscript. Please check Page 15 of the revised manuscript (Figure 4 and the highlighted text)) for details. The result is consistent with the main message but contains more information.
>
> Intuitively, the reason why PAC-Bayes training is more stable to hyperparameters is that it directly minimizes a relatively tight bound on the test error, rather than minimizing the training error as in ERM, eliminating the need for additional regularizations. As a result, hyperparameters in PAC-Bayes primarily control the convergence rate (or whether it converges at all) rather than the regularization strength, as in ERM.  The additional trainable model parameters in PAC-Bayes also mostly increase the difficulty of convergence (and memory) but do not much affect the final generalization.
>
> Based on other reviewers' suggestions, the revised version also includes the following changes:
> 1) We added a motivation section (Sec. 4.1) to illustrate why the previous PAC-Bayes bounds are insufficient, thus motivating our work.
> 2) We modified and shortened the contribution statement (page 2) to highlight only the main contribution.
> 3) We added Remark 4.12 and Figures 1 and 2 to explain where the improvement of our new bound comes from and how much tighter it is numerically compared to existing bounds.
> 4) We modified the titles of sections and subsections to make the structure clearer.

---

### Decision · Action_Editor_K5Zh · 2024-09-03

**Recommendation:** Accept with minor revision

**Comment:**

The authors have carefully revised their work following the reviews and discussion with reviewers, and this submission is now worthy of publication at TMLR. I kindly invite the authors to polish the last details and remove the colours used to signal updates from the initial submission. Congratulations on a fine piece of work!

**Audience:**

There has been a surge of interest in PAC-Bayes learning over the past decade and this paper adds to a rich literature on the topic, including papers already published at TMLR. I am confident this paper will be of interest to some of the TMLR's audience.

**Claims And Evidence:**

All claims (theoretical and empirical) are appropriately supported by evidence.

---

> ### Author Response · Authors · 2024-09-18
>
> Dear AE,
>
> We have completed the final polishing and removed the color. Please let us know if you have any additional suggestions!
>
> Thank you very much for your time and feedback!

---

> > ### Comment · Action_Editor_K5Zh · 2024-09-19
> >
> > Just had a look and this looks fine to me -- you should also upload a de-anonymised version of your paper. I believe you should use the upload camera-ready option -- until you do we can't properly validate.
> >
> > Best,
> >
> > Benjamin

---

> > > ### Author Response · Authors · 2024-09-24
> > > **Camera Ready Revision**
> > >
> > > Dear AE,
> > >
> > > We have submitted the camera-ready version, including the author's information, acknowledgment, and an additional citation [1] in the related work section.
> > >
> > > Thank you for your valuable feedback!
> > >
> > > [1] Viallard, Paul, et al. "Learning via Wasserstein-based high probability generalisation bounds." Advances in Neural Information Processing Systems 36 (2024).

---

> > > > ### Comment · Action_Editor_K5Zh · 2024-09-27
> > > >
> > > > Dear authors,
> > > >
> > > > Thanks -- can you please update that last reference as it appeared at NeurIPS 2023, not 2024?
> > > >
> > > > Thanks

---

> > > > > ### Author Response · Authors · 2024-09-27
> > > > >
> > > > > Dear AE,
> > > > >
> > > > > Thank you for pointing this out! We've updated the reference to NeurIPS 2023.